# Zero-Sum Stochastic Stackelberg Games

**Denizalp Goktas**
Department of Computer Science
Brown University
Providence, RI 02906, USA
denizalp_goktas@brown.edu

**Jiayi Zhao**
Department of Computer Science
Pomona College
Pomona, CA, USA
jzae2019@mymail.pomona.edu

**Amy Greenwald**
Brown University
Providence, RI 02906, USA
amy_greenwald@brown.edu

## Abstract

Zero-sum stochastic games have found important applications in a variety of fields, from machine learning to economics. Work on this model has primarily focused on the computation of Nash equilibrium due to its effectiveness in solving adversarial board and video games. Unfortunately, a Nash equilibrium is not guaranteed to exist in zero-sum stochastic games when the payoffs at each state are not convex-concave in the players' actions. A Stackelberg equilibrium, however, is guaranteed to exist. Consequently, in this paper, we study zero-sum stochastic Stackelberg games. Going beyond known existence results for (non-stationary) Stackelberg equilibria, we prove the existence of recursive (i.e., Markov perfect) Stackelberg equilibria (recSE) in these games, provide necessary and sufficient conditions for a policy profile to be a recSE, and show that recSE can be computed in (weakly) polynomial time via value iteration. Finally, we show that zero-sum stochastic Stackelberg games can model the problem of pricing and allocating goods across agents and time. More specifically, we propose a zero-sum stochastic Stackelberg game whose recSE correspond to the recursive competitive equilibria of a large class of stochastic Fisher markets. We close with a series of experiments that showcase how our methodology can be used to solve the consumption-savings problem in stochastic Fisher markets.

Min-max optimization has paved the way for recent progress in a variety of fields, from machine learning to economics. These applications require computing solutions to a **constrained min-max optimization problem** i.e., $\min_{\boldsymbol{x} \in \mathcal{X}} \max_{\boldsymbol{y} \in \mathcal{Y}} f(\boldsymbol{x}, \boldsymbol{y})$, where the objective function $f : \mathcal{X} \times \mathcal{Y} \to \mathbb{R}$ is continuous, and the constraint sets $\mathcal{X} \subset \mathbb{R}^n$ and $\mathcal{Y} \subset \mathbb{R}^m$ are nonempty and compact. When $f$ is convex-concave, and the constraint sets $\mathcal{X}$ and $\mathcal{Y}$ are convex, the seminal minimax theorem [1, 2] holds, i.e., $\min_{\boldsymbol{x} \in \mathcal{X}} \max_{\boldsymbol{y} \in \mathcal{Y}} f(\boldsymbol{x}, \boldsymbol{y}) = \max_{\boldsymbol{y} \in \mathcal{Y}} \min_{\boldsymbol{x} \in \mathcal{X}} f(\boldsymbol{x}, \boldsymbol{y})$, and such problems can be interpreted as computing a Nash equilibrium of a simultaneous-move **min-max** (or **zero-sum**) **game** between an **outer player** $\boldsymbol{x}$ and an **inner player** $\boldsymbol{y}$ with respective payoff functions $-f$, $f$ and respective action sets $\mathcal{X}, \mathcal{Y}$, where the solutions $(\boldsymbol{x}^*, \boldsymbol{y}^*) \in \mathcal{X} \times \mathcal{Y}$ are best responses to one another.

More generally, one can consider **zero-sum stochastic games**, played over an infinite discrete time horizon $\mathbb{N}_+$. The game starts at some initial state $\boldsymbol{S}^{(0)} \sim \mu^{(0)}$. At each subsequent time-step $t \in \mathbb{N}_+$, players encounter a new state $\boldsymbol{s}^{(t)} \in \mathcal{S}$. After taking their respective actions $(\boldsymbol{x}^{(t)}, \boldsymbol{y}^{(t)})$ from their respective action spaces $\mathcal{X}(\boldsymbol{s}^{(t)}) \subseteq \mathbb{R}^n$ and $\mathcal{Y}(\boldsymbol{s}^{(t)}) \subseteq \mathbb{R}^m$, they receive payoffs $r(\boldsymbol{s}^{(t)}, \boldsymbol{x}^{(t)}, \boldsymbol{y}^{(t)})$, and then either transition to a new state $\boldsymbol{S}^{(t+1)} \sim p(\cdot \mid \boldsymbol{s}^{(t)}, \boldsymbol{x}^{(t)}, \boldsymbol{y}^{(t)})$ with probability $\gamma$, or the game ends with the remaining probability. The goal of the outer (resp. inner) player is to play a sequence

36th Conference on Neural Information Processing Systems (NeurIPS 2022).

of actions $\{\boldsymbol{x}^{(t)}\}_t$ (resp. $\{\boldsymbol{y}^{(t)}\}_t$), that maximizes (resp. minimizes) their expected cumulative discounted payoff (resp. loss) $\mathbb{E}\left[\sum_{t=0}^{\infty}\gamma^t r(\boldsymbol{S}^{(t)}, \boldsymbol{x}^{(t)}, \boldsymbol{y}^{(t)})\right]$, fixing their opponent's policy.

A **(stationary) policy** is a mapping from states to actions. When $r(\boldsymbol{s}, \boldsymbol{x}, \boldsymbol{y})$ is bounded, continuous, and concave-convex in $(\boldsymbol{x}, \boldsymbol{y})$, for all $\boldsymbol{s} \in \mathcal{S}$, we are guaranteed the existence of a stationary **policy profile**, i.e., a pair of policies $\boldsymbol{\pi_x} : \mathcal{S} \to \mathcal{X}$, $\boldsymbol{\pi_y} : \mathcal{S} \to \mathcal{Y}$ for the outer and inner players, respectively, specifying the actions taken at each state, with a unique value such that both players maximize their expected payoffs, as a generalization of the minimax theorem holds [3]:[1]

$$\min_{\boldsymbol{\pi_x} \in \mathcal{X}^{\mathcal{S}}} \max_{\boldsymbol{\pi_y} \in \mathcal{Y}^{\mathcal{S}}} \mathbb{E}\left[\sum_{t=0}^{\infty} \gamma^t r(\boldsymbol{S}^{(t)}, \boldsymbol{\pi_x}(\boldsymbol{S}^{(t)}), \boldsymbol{\pi_y}(\boldsymbol{S}^{(t)}))\right] = \max_{\boldsymbol{\pi_y} \in \mathcal{Y}^{\mathcal{S}}} \min_{\boldsymbol{\pi_x} \in \mathcal{X}^{\mathcal{S}}} \mathbb{E}\left[\sum_{t=0}^{\infty} \gamma^t r(\boldsymbol{S}^{(t)}, \boldsymbol{\pi_x}(\boldsymbol{S}^{(t)}), \boldsymbol{\pi_y}(\boldsymbol{S}^{(t)}))\right]$$

In other words, under the aforementioned assumptions, we are guaranteed the existence of a **recursive Nash equilibrium** (sometimes called a Markov perfect Nash equilibrium [4]), a stationary policy profile in which players not only best respond to one another, but they do so at every state of the game. Additionally, when the rewards at each state are convex-concave, a recursive Nash equilibrium can be computed in polynomial time by iterative application of the min-max operator [3]. Zero-sum *stochastic* games generalize zero-sum games from a single state to multiple states, and have found even more applications in a variety of fields [5].

Unfortunately, when the objective function in a min-max optimization problem is not convex-concave, a minimax theorem is not guaranteed to hold, precluding the interpretation of the game as simultaneous-move, and the guaranteed existence of Nash equilibrium. Nonetheless, the game can still be viewed as a Stackelberg game, in which the outer player moves before the inner one. The canonical solution concept in such games is **Stackelberg equilibrium (SE)**. Moreover, in Stackelberg games, the inner player's actions can be constrained by the outer player's choice, without impacting existence [6]. The result is a **min-max Stackelberg game**: i.e., $\min_{\boldsymbol{x} \in \mathcal{X}} \max_{\boldsymbol{y} \in \mathcal{Y}: h(\boldsymbol{x}, \boldsymbol{y}) \geq 0} f(\boldsymbol{x}, \boldsymbol{y})$ where $f, h : \mathcal{X} \times \mathcal{Y} \to \mathbb{R}$ are continuous, and $\mathcal{X}, \mathcal{Y}$ are non-empty and compact. Even more problems of interest can be captured by this model [7, 8, 9, 10].

One can likewise consider **zero-sum stochastic Stackelberg games**, which generalize both zero-sum Stackelberg games and zero-sum stochastic games. Similar to zero-sum stochastic games, these games are played over an infinite discrete time horizon $\mathbb{N}_+$, start at some state $\boldsymbol{S}^{(0)} \sim \mu^{(0)}$ and consist of nonempty and compact actions spaces $\mathcal{X} \subset \mathbb{R}^n$ and $\mathcal{Y} \subset \mathbb{R}^m$[2], a state-dependent payoff function $r(\boldsymbol{s}, \boldsymbol{x}, \boldsymbol{y})$, a transition probability $p(\boldsymbol{s}' \mid \boldsymbol{s}, \boldsymbol{x}, \boldsymbol{y})$, and a discount rate $\gamma$, but are in addition augmented with a state-dependent (joint action) constraint function $\boldsymbol{g}(\boldsymbol{s}, \boldsymbol{x}, \boldsymbol{y})$, with two players that seek to optimize their cumulative discounted payoffs, in expectation, while satisfying the constraint $\boldsymbol{g}(\boldsymbol{s}, \boldsymbol{x}, \boldsymbol{y}) \geq \boldsymbol{0}$ at each state $\boldsymbol{s} \in \mathcal{S}$. Applications of this model include autonomous driving [8, 10], reach-avoid problems in human-robot interaction [9], and robust optimization in stochastic environments [7], and, as we show, economic markets.

While in stochastic games, players announce their policies simultaneously before play commences, in stochastic Stackelberg games, the outer player, announces their (in general, non-stationary) policy: i.e., the action they will take at each time step, first, after which the inner player announces theirs. The canonical solution concept for such games is the **Stackelberg equilibrium**, which is guaranteed to exist (in non-stationary policies) under mild assumptions [11, 6].

The computational complexity of non-stationary equilibrium policies in stochastic games can be prohibitive, since even representing such policies in an infinite horizon setting is intractable. A natural question to ask then is whether *stationary* equilibria exist in zero-sum Stackelberg games, i.e., stationary policy profiles at which the outer player maximizes their expected discounted cumulative payoff while the inner player best responds. We call such policies **recursive Stackelberg equilibria (recSE)** (or Markov perfect Stackelberg equilibria).

In this paper, we define and prove the existence of recSE in zero-sum stochastic Stackelberg games, provide necessary and sufficient conditions for a policy profile to be a recSE, and show that a recSE can be computed in (weakly) polynomial time via value iteration. We further show that zero-sum stochastic Stackelberg games can be used to solve problems of pricing and allocating goods across

---

[1]Shapley's original results, which concern state-dependent payoff functions that are bilinear in the outer and inner players' actions, extend directly to payoffs which are convex-concave in the players' actions.

[2]To simplify notation, we drop the dependency of action spaces on states going forward, but our theory applies in this more general setting.

agents and time. In particular, we introduce **stochastic Fisher markets**, a stochastic generalization of the Fisher market [12], and a special case of Friesen's [13] financial market model, which itself is a stochastic generalization of the Arrow and Debreu model of a competitive economy [14]. We then prove the existence of recursive competitive equilibrium (recCE) [15] in this model, under the assumption that consumers have continuous and homogeneous utility functions, by characterizing the recCE of any stochastic Fisher market as the recSE of a corresponding zero-sum stochastic Stackelberg game. Finally, we use value iteration to solve various stochastic Fisher markets, highlighting the issues that value iteration can encounter, depending on the smoothness properties of the utilities.

**Related Work**    Algorithms for min-max optimization problems (i.e., zero-sum games) with independent strategy sets have been extensively studied [16, 17, 18, 19, 20, 21, 22, 23, 24, 25, 26, 27, 28, 29, 30, 31, 32, 33, 34, 35, 36, 37, 38, 39] (for a summary see, Section G [6]). Goktas and Greenwald studied min-max games with dependent strategy sets, proposing polynomial-time nested gradient descent ascent (GDA) [6] and simultaneous GDA algorithms for such problems [40].

The computation of Stackelberg equilibrium in two-player stochastic Stackelberg games has been studied in several interesting settings, in which the leader moves before the follower, but without the leader's actions impacting the followers' choice sets. Bensoussan, Chen, and Sethi [41] study continuous-time general-sum stochastic Stackelberg games with continuous action spaces, and prove existence of a solution in this setting. Vorobeychik and Singh [11] consider a general-sum stochastic Stackelberg game with finite state-action spaces and an infinite horizon. These authors show that stationary SE policies do not exist in this very general setting, but nonetheless identify a subclass of games, namely team (or potential) Stackelberg games for which stationary Stackelberg equilibrium policies do exist. Vu et al. [42] study the empirical convergence of policy gradient methods in the same setting as Vorobeychik and Singh [11], while Ramponi and Restelli [43] study non-stationary equilibria in this same setting, assuming a finite horizon. Chang, Erera, and White [44] and Sengupta and Kambhampati [45] consider a partially observable version of Vorobeychik and Singh's [11] model, and provide methods to compute Stackelberg equilibria in their setting.

Some recent research concerns one leader-many followers Stackelberg games. Vasal [46] studies a discrete-time, finite horizon one leader-many follower stochastic Stackelberg game with discrete action and state spaces, and provides algorithms to solve such games. DeMiguel and Xu [47] consider a stochastic Stackelberg game-like market model with $n$ leaders and $m$ followers; they prove the existence of a SE in their model, and provide (without theoretical guarantees) algorithms that converge to such an equilibrium in experiments. Dynamic Stackelberg games [48] have been applied to a wide range of problems, including security [46, 11], insurance provision [49, 50], advertising [51], robust agent design [52], allocating goods across time intertemporal pricing [53].

The study of algorithms that compute competitive equilibria in Fisher markets was initiated by Devanur et al., who provided a polynomial-time method for solving these markets assuming linear utilities. More recently, there have been efforts to study markets in dynamic settings [55, 56, 6], in which the goal is to either track the changing equilibrium of a changing market, or minimize some regret-like quantity for the market. The models considered in these earlier works differ from ours as they do not have stochastic structure and do not invoke a dynamic solution concept.

## 1   Preliminaries

**Notation**    We use caligraphic uppercase letters to denote sets (e.g., $\mathcal{X}$); bold lowercase letters to denote vectors (e.g., $\boldsymbol{p}, \boldsymbol{\pi}$); bold uppercase letters to denote matrices and vector-valued random variables (e.g., $\boldsymbol{X}, \boldsymbol{\Gamma}$)—which one should be clear from context; lowercase letters to denote scalar quantities (e.g., $x, \gamma$); and uppercase letters to denote scalar-valued random variables (e.g., $X, \Gamma$). We denote the $i$th row vector of a matrix (e.g., $\boldsymbol{X}$) by the corresponding bold lowercase letter with subscript $i$ (e.g., $\boldsymbol{x}_i$). Similarly, we denote the $j$th entry of a vector (e.g., $\boldsymbol{p}$ or $\boldsymbol{x}_i$) by the corresponding Roman lowercase letter with subscript $j$ (e.g., $p_j$ or $x_{ij}$). We denote functions by a letter: e.g., $f$ if the function is scalar valued, and $\boldsymbol{f}$ if the function is vector valued. We denote the vector of ones of size $n$ by $\boldsymbol{1}_n$. We denote the set of integers $\{1, \ldots, n\}$ by $[n]$, the set of natural numbers by $\mathbb{N}$, the set of real numbers by $\mathbb{R}$. We denote the postive and strictly positive elements of a set by a $+$ and $++$ subscript respectively, e.g., $\mathbb{R}_+$ and $\mathbb{R}_{++}$. We denote the orthogonal projection operator onto a set $C$ by $\Pi_C$, i.e., $\Pi_C(\boldsymbol{x}) = \arg\min_{\boldsymbol{y} \in C} \|\boldsymbol{x} - \boldsymbol{y}\|^2$. We denote by $\Delta_n = \{\boldsymbol{x} \in \mathbb{R}_+^n \mid \sum_{i=1}^n x_i = 1\}$, and by $\Delta(A)$, the set of probability measures on the set $A$.

A **stochastic Stackelberg game** $(\mathcal{S}, \mathcal{X}, \mathcal{Y}, \mu^{(0)}, r_{\boldsymbol{x}}, r_{\boldsymbol{y}}, \boldsymbol{g}, p, \gamma)$ is a two-player game played over an infinite discrete time horizon $\mathbb{N}_+$. At each time-step $t \in \mathbb{N}_+$, the players, who we call the outer- (resp. inner-) players, encounter a new state $\boldsymbol{s} \in \mathcal{S}$, and choose an action to play from their continuous set of actions $\mathcal{X} \subset \mathbb{R}^n$ (resp. $\mathcal{Y} \subset \mathbb{R}^m$). Play initiates at a start state $\boldsymbol{S}^{(0)}$ drawn from a distribution $\mu^{(0)} : \mathcal{S} \to [0,1]$. At each state $\boldsymbol{s} \in \mathcal{S}$ the action $\boldsymbol{x} \in \mathcal{X}$ chosen by the outer player determines the set of **feasible** actions $\{\boldsymbol{y} \in \mathcal{Y} \mid \boldsymbol{g}(\boldsymbol{s}, \boldsymbol{x}, \boldsymbol{y}) \geq \boldsymbol{0}\}$ available to the inner player, where $\boldsymbol{g} : \mathcal{S} \times \mathcal{X} \times \mathcal{Y} \to \mathbb{R}^d$. After the outer and inner players both make their moves, they receive payoffs $r_{\boldsymbol{x}} : \mathcal{S} \times \mathcal{X} \times \mathcal{Y} \to \mathbb{R}$ and $r_{\boldsymbol{y}} : \mathcal{S} \times \mathcal{X} \times \mathcal{Y} \to \mathbb{R}$, respectively, and the game either ends with probability $1 - \gamma$, where $\gamma \in (0,1)$ is called the **discount factor**, or transitions to a new state $\boldsymbol{s}' \in \mathcal{S}$, according to a **transition** probability function $p : \mathcal{S} \times \mathcal{S} \times \mathcal{X} \times \mathcal{Y} \to [0,1]$ s.t. $p(\boldsymbol{s}' \mid \boldsymbol{s}, \boldsymbol{x}, \boldsymbol{y}) \in [0,1]$ denotes the probability of transitioning to state $\boldsymbol{s}' \in \mathcal{S}$ from state $\boldsymbol{s} \in \mathcal{S}$ when action profile $(\boldsymbol{x}, \boldsymbol{y}) \in \mathcal{X} \times \mathcal{Y}$ is chosen by the players.

In this paper, we focus on **zero-sum** stochastic Stackelberg games $\mathcal{G}^{(0)} \doteq (\mathcal{S}, \mathcal{X}, \mathcal{Y}, \mu^{(0)}, r, \boldsymbol{g}, p, \gamma)$, in which the outer player's loss is the inner player's gain, i.e., $r_{\boldsymbol{x}} = -r_{\boldsymbol{y}}$. A zero-sum stochastic Stackelberg game reduces to zero-sum (simultaneous-move) stochastic game [3] in the special case where $\boldsymbol{g}(\boldsymbol{s}, \boldsymbol{x}, \boldsymbol{y}) \geq 0$, for all state-action tuples $(\boldsymbol{s}, \boldsymbol{x}, \boldsymbol{y}) \in \mathcal{S} \times \mathcal{X} \times \mathcal{Y}$. More generally, a policy profile $(\boldsymbol{\pi_x}, \boldsymbol{\pi_y}) \in \mathcal{X}^{\mathcal{S}} \times \mathcal{Y}^{\mathcal{S}}$ is said to be **feasible** if $\boldsymbol{g}(\boldsymbol{s}, \boldsymbol{\pi_x}(\boldsymbol{s}), \boldsymbol{\pi_y}(\boldsymbol{s})) \geq 0$, for all states $\boldsymbol{s} \in \mathcal{S}$. To simplify notation, we introduce a function $\boldsymbol{G} : \mathcal{X}^{\mathcal{S}} \times \mathcal{Y}^{\mathcal{S}} \to \mathbb{R}^{|\mathcal{S}| \times d}$ such that $\boldsymbol{G}(\boldsymbol{\pi_x}, \boldsymbol{\pi_y}) = (\boldsymbol{g}(\boldsymbol{s}, \boldsymbol{\pi_x}(\boldsymbol{s}), \boldsymbol{\pi_y}(\boldsymbol{s})))_{\boldsymbol{s} \in \mathcal{S}}$, and define feasible policy profiles as those $(\boldsymbol{\pi_x}, \boldsymbol{\pi_y}) \in \mathcal{X}^{\mathcal{S}} \times \mathcal{Y}^{\mathcal{S}}$ s.t. $\boldsymbol{G}(\boldsymbol{\pi_x}, \boldsymbol{\pi_y}) \geq \boldsymbol{0}$. From now on, we assume:

**Assumption 1.1.** *1. For all states $\boldsymbol{s} \in \mathcal{S}$, the functions $r(\boldsymbol{s}, \cdot, \cdot), \boldsymbol{g}(\boldsymbol{s}, \cdot, \cdot)$ are continuous in $(\boldsymbol{x}, \boldsymbol{y}) \in \mathcal{X} \times \mathcal{Y}$, and payoffs are bounded, i.e., $\|r\|_{\infty} \leq r_{\max} < \infty$, for some $r_{\max} \in \mathbb{R}_+$, 2. $\mathcal{X}, \mathcal{Y}$ are non-empty and compact, and for all $\boldsymbol{s} \in \mathcal{S}$ and $\boldsymbol{x} \in \mathcal{X}$ there exists $\boldsymbol{y} \in \mathcal{Y}$ s.t. $\boldsymbol{g}(\boldsymbol{s}, \boldsymbol{x}, \boldsymbol{y}) \geq \boldsymbol{0}$.*[3]

Given a zero-sum stochastic Stackelberg game $\mathcal{G}^{(0)}$, the **state-value function**, $v : \mathcal{S} \times \mathcal{X}^{\mathcal{S}} \times \mathcal{Y}^{\mathcal{S}} \to \mathbb{R}$, and the **action-value function**, $q : \mathcal{S} \times \mathcal{X} \times \mathcal{Y} \times \mathcal{X}^{\mathcal{S}} \times \mathcal{Y}^{\mathcal{S}} \to \mathbb{R}$, respectively, are defined as:

$$v(\boldsymbol{s}; \boldsymbol{\pi_x}, \boldsymbol{\pi_y}) = \mathbb{E}^{\boldsymbol{\pi_x}, \boldsymbol{\pi_y}}_{\boldsymbol{S}^{(t+1)} \sim p(\cdot \mid \boldsymbol{S}^{(t)}, \boldsymbol{X}^{(t)}, \boldsymbol{Y}^{(t)})} \left[ \sum_{t=0}^{\infty} \gamma^t r(\boldsymbol{S}^{(t)}, \boldsymbol{X}^{(t)}, \boldsymbol{Y}^{(t)}) \mid \boldsymbol{S}^{(0)} = \boldsymbol{s} \right] \tag{1}$$

$$q(\boldsymbol{s}, \boldsymbol{x}, \boldsymbol{y}; \boldsymbol{\pi_x}, \boldsymbol{\pi_y}) = \mathbb{E}^{\boldsymbol{\pi_x}, \boldsymbol{\pi_y}}_{\boldsymbol{S}^{(t+1)} \sim p(\cdot \mid \boldsymbol{S}^{(t)}, \boldsymbol{X}^{(t)}, \boldsymbol{Y}^{(t)})} \left[ \sum_{t=0}^{\infty} \gamma^t r(\boldsymbol{S}^{(t)}, \boldsymbol{X}^{(t)}, \boldsymbol{Y}^{(t)}) \mid \boldsymbol{S}^{(0)} = \boldsymbol{s}, \boldsymbol{X}^{(0)} = \boldsymbol{x}, \boldsymbol{Y}^{(0)} = \boldsymbol{y} \right]$$
$$\tag{2}$$

Again, to simplify notation, we write expectations conditional on $\boldsymbol{X}^{(t)} = \boldsymbol{\pi_x}(\boldsymbol{S}^{(t)})$ and $\boldsymbol{Y}^{(t)} = \boldsymbol{\pi_y}(\boldsymbol{S}^{(t)})$ as $\mathbb{E}^{\boldsymbol{\pi_x}, \boldsymbol{\pi_y}}$, and denote the state- and action-value functions by $v^{\boldsymbol{\pi_x}\boldsymbol{\pi_y}}(\boldsymbol{s})$, and $q^{\boldsymbol{\pi_x}\boldsymbol{\pi_y}}(\boldsymbol{s}, \boldsymbol{x}, \boldsymbol{y})$, respectively. Additionally, we let $\mathcal{V} = [-r_{\max}/1-\gamma, r_{\max}/1-\gamma]^{\mathcal{S}}$ be the space of all state-value functions of the form $v : \mathcal{S} \to [-r_{\max}/1-\gamma, r_{\max}/1-\gamma]$, and we let $\mathcal{Q} = [-r_{\max}/1-\gamma, r_{\max}/1-\gamma]^{\mathcal{S} \times \mathcal{X} \times \mathcal{Y}}$ be the space of all action-value functions of the form $q : \mathcal{S} \times \mathcal{X} \times \mathcal{Y} \to [-r_{\max}/1-\gamma, r_{\max}/1-\gamma]$. Note that by Assumption 1.1 the range of the state- and action-value functions is $[-r_{\max}/1-\gamma, r_{\max}/1-\gamma]$. The cumulative payoff function of the game $u : \mathcal{X}^{\mathcal{S}} \times \mathcal{Y}^{\mathcal{S}} \to \mathbb{R}$ is the total expected loss (resp. gain) of the outer (resp. inner) player, given by $u(\boldsymbol{\pi_x}, \boldsymbol{\pi_y}) = \mathbb{E}_{\boldsymbol{s} \sim \mu^{(0)}(\boldsymbol{s})}[v^{\boldsymbol{\pi_x}\boldsymbol{\pi_y}}(\boldsymbol{s})]$.

The canonical solution concept for stochastic Stackelberg games is the **Stackelberg equilibrium** **(SE)**. A feasible policy profile $(\boldsymbol{\pi_x^*}, \boldsymbol{\pi_y^*}) \in \mathcal{X}^{\mathcal{S}} \times \mathcal{Y}^{\mathcal{S}}$ is said to be a Stackelberg equilibrium (SE) of a zero-sum stochastic Stackelberg game $\mathcal{G}^{(0)}$ iff

$$\max_{\boldsymbol{\pi_y} \in \mathcal{Y}^{\mathcal{S}} : \boldsymbol{G}(\boldsymbol{\pi_x^*}, \boldsymbol{\pi_y}) \geq \boldsymbol{0}} u\left(\boldsymbol{\pi_x^*}, \boldsymbol{\pi_y}\right) \leq u\left(\boldsymbol{\pi_x^*}, \boldsymbol{\pi_y^*}\right) \leq \min_{\boldsymbol{\pi_x} \in \mathcal{X}^{\mathcal{S}}} \max_{\boldsymbol{\pi_y} \in \mathcal{Y}^{\mathcal{S}} : \boldsymbol{G}(\boldsymbol{\pi_x}, \boldsymbol{\pi_y}) \geq \boldsymbol{0}} u\left(\boldsymbol{\pi_x}, \boldsymbol{\pi_y}\right) \ .$$

Note the strength of this definition, as it requires the constraints $\boldsymbol{g}(\boldsymbol{s}, \boldsymbol{\pi_x}, \boldsymbol{\pi_y}) \geq \boldsymbol{0}$ to be satisfied at all states $\boldsymbol{s} \in \mathcal{S}$, not only states which are reached with strictly positive probability. A SE is

---

[3]Note that this condition is weaker than Slater's condition; it simply ensures the feasible action sets are non-empty for the inner player at each state.

guaranteed to exist in zero-sum stochastic Stackelberg games, under Assumption 1.1, as a corollary of Goktas and Greenwald's [6] Proposition B.2.; however, this existence result is non-constructive.[4]

In this paper, we study a Markov perfect refinement of SE, which we call **recursive Stackelberg equilibrium** (recSE).

**Definition 1.2** (Recursive Stackelberg Equilibrium (recSE)). *A policy profile $(\pi_x^*, \pi_y^*) \in \mathcal{S}^{\mathcal{X}} \times \mathcal{S}^{\mathcal{Y}}$ is a **recursive Stackelberg equilibrium (recSE)** iff, for all $s \in \mathcal{S}$, it holds that:*

$$\max_{\boldsymbol{y} \in \mathcal{Y}: g(s, \pi_x^*(s), \boldsymbol{y}) \geq \mathbf{0}} q^{\pi_x^* \pi_y^*}(s, \pi_x^*(s), \boldsymbol{y}) \leq q^{\pi_x^* \pi_y^*}(s, \pi_x^*(\boldsymbol{x}), \pi_y^*(\boldsymbol{y})) \leq \min_{\boldsymbol{x} \in \mathcal{X}} \max_{\boldsymbol{y} \in \mathcal{Y}: g(s, \boldsymbol{x}, \boldsymbol{y}) \geq \mathbf{0}} q^{\pi_x^* \pi_y^*}(s, \boldsymbol{x}, \boldsymbol{y}).$$

Equivalently, a policy profile $(\pi_x^*, \pi_y^*)$ is a recSE if $(\pi_x^*(s), \pi_y^*(s))$ is a SE with value $v^{\pi_x^* \pi_y^*}(s)$ at each state $s \in \mathcal{S}$: i.e., $v^{\pi_x^* \pi_y^*}(s) = \min_{\boldsymbol{x} \in \mathcal{X}} \max_{\boldsymbol{y} \in \mathcal{Y}: g(s, \boldsymbol{x}, \boldsymbol{y}) \geq \mathbf{0}} q^{\pi_x^* \pi_y^*}(s, \boldsymbol{x}, \boldsymbol{y})$, for all $s \in \mathcal{S}$.

**Mathematical Preliminaries** A probability measure $q_1 \in \Delta(\mathcal{S})$ **convex stochastically dominates (CSD)** $q_2 \in \Delta(\mathcal{S})$ if $\int_{\mathcal{S}} v(s) q_1(s) ds \geq \int_{\mathcal{S}} v(s) q_2(s) ds$ for all continuous, bounded, and convex functions $v$ on $S$. A transition function $p$ is termed **CSD convex** in $\boldsymbol{x}$ if, for all $\lambda \in (0, 1)$, $\boldsymbol{y} \in \mathcal{Y}$ and any $(s', \boldsymbol{x}'), (s^\dagger, \boldsymbol{x}^\dagger) \in \mathcal{S} \times \mathcal{X}$, with $(s, \boldsymbol{x}) = \lambda(s', \boldsymbol{x}') + (1 - \lambda)(s^\dagger, \boldsymbol{x}^\dagger)$, it holds that $\lambda p(\cdot \mid s', \boldsymbol{x}', \boldsymbol{y}) + (1 - \lambda) p(\cdot \mid s^\dagger, \boldsymbol{x}^\dagger, \boldsymbol{y})$ CSD $p(\cdot \mid s, \boldsymbol{x}, \boldsymbol{y})$. A transition function $p$ is termed **CSD concave** in $\boldsymbol{y}$ if, for all $\lambda \in (0, 1)$ and any $(s', \boldsymbol{y}'), (s^\dagger, \boldsymbol{y}^\dagger) \in \mathcal{S} \times \mathcal{X} \times \mathcal{Y}$, with $(s, \boldsymbol{y}) = \lambda(s', \boldsymbol{y}') + (1 - \lambda)(s^\dagger, \boldsymbol{y}^\dagger)$, it holds that $p(\cdot \mid s, \boldsymbol{x}, \boldsymbol{y})$ CSD $\lambda p(\cdot \mid s', \boldsymbol{x}, \boldsymbol{y}') + (1 - \lambda) p(\cdot \mid s^\dagger, \boldsymbol{x}, \boldsymbol{y}^\dagger)$. A mapping $L : \mathcal{A} \to \mathcal{B}$ is said to be a **contraction mapping** (resp. **non-expansion**) w.r.t. norm $\|\cdot\|$ iff for all $\boldsymbol{x}, \boldsymbol{y} \in \mathcal{A}$, and for $k \in [0, 1)$ (resp. $k = 1$) such that $\|L(\boldsymbol{x}) - L(\boldsymbol{y})\| \leq k \|\boldsymbol{x} - \boldsymbol{y}\|$. The **min-max operator** $\min_{\boldsymbol{x} \in \mathcal{X}} \max_{\boldsymbol{y} \in \mathcal{Y}} : \mathbb{R}^{\mathcal{X} \times \mathcal{Y}} \to \mathbb{R}$ w.r.t. to sets $\mathcal{X}, \mathcal{Y}$ takes as input a function $f : \mathcal{X} \times \mathcal{Y} \to \mathbb{R}$ and outputs $\min_{\boldsymbol{x} \in \mathcal{X}} \max_{\boldsymbol{y} \in \mathcal{Y}} f(\boldsymbol{x}, \boldsymbol{y})$. The **generalized min-max operator** $\min_{\boldsymbol{x} \in \mathcal{X}} \max_{\boldsymbol{y} \in \mathcal{Y}: g(\boldsymbol{x}, \boldsymbol{y}) \geq \mathbf{0}} : \mathbb{R}^{\mathcal{X} \times \mathcal{Y}} \to \mathbb{R}$ w.r.t. to sets $\mathcal{X}, \mathcal{Y}$ and the function $\boldsymbol{g} : \mathcal{X} \times \mathcal{Y} \to \mathbb{R}$ takes as input a function $f : \mathcal{X} \times \mathcal{Y} \to \mathbb{R}$ and outputs $\min_{\boldsymbol{x} \in \mathcal{X}} \max_{\boldsymbol{y} \in \mathcal{Y}: g(\boldsymbol{x}, \boldsymbol{y}) \geq \mathbf{0}} f(\boldsymbol{x}, \boldsymbol{y})$.

## 2 Properties of Recursive Stackelberg equilibrium

In this section, we show that a recSE exists in all zero-sum stochastic Stackelberg games.[5] To do so, we first associate an operator $C : \mathcal{V} \to \mathcal{V}$ with any zero-sum stochastic Stackelberg game $\mathcal{G}^{(0)}$, the fixed points of which satisfy Definition 1.2, and hence correspond to the value function associated with a recSE of $\mathcal{G}^{(0)}$. We then show that this operator is a contraction mapping, thereby establishing the existence of such a fixed point. This result generalizes Shapley's theorem on the existence of Markov perfect Nash equilibria in zero-sum stochastic games [3]. Define $C : \mathcal{V} \to \mathcal{V}$ for any zero-sum stochastic Stackelberg game $\mathcal{G}^{(0)}$ as the operator $(Cv)(s) = \min_{\boldsymbol{x} \in \mathcal{X}} \max_{\boldsymbol{y} \in \mathcal{Y}: g(s, \boldsymbol{x}, \boldsymbol{y}) \geq \mathbf{0}} \mathbb{E}_{\boldsymbol{S}' \sim p(\cdot | s, \boldsymbol{x}, \boldsymbol{y})} [r(s, \boldsymbol{x}, \boldsymbol{y}) + \gamma v(\boldsymbol{S}')]$. We first show that the fixed points of $C$ correspond to the recSE of the associated game.

**Theorem 2.1.** *$(\pi_x^*, \pi_y^*)$ is a recSE of $\mathcal{G}^{(0)}$ of $v^{\pi_x \pi_y}$ iff it induces a value function which is a fixed point of $C$: i.e., $(\pi_x^*, \pi_y^*)$ is a Stackelberg equilbrium iff, for all $s \in \mathcal{S}$, $\left( C v^{\pi_x^* \pi_y^*} \right)(s) = v^{\pi_x^* \pi_y^*}(s)$.*

The following technical lemma is crucial to proving that $C$ is a contraction mapping. It tells us that the generalized min-max operator is non-expansive; in other words, the generalized min-max operator is 1-Lipschitz w.r.t. the sup-norm.

**Lemma 2.2.** *Suppose that $f, h : \mathcal{X} \times \mathcal{Y} \to \mathbb{R}$, $\boldsymbol{g} : \mathcal{X} \times \mathcal{Y} \to \mathbb{R}^d$ are continuous functions, and $\mathcal{X}, \mathcal{Y}$ are compact sets. Then $\left| \min_{\boldsymbol{x} \in \mathcal{X}} \max_{\boldsymbol{y} \in \mathcal{Y}: g(\boldsymbol{x}, \boldsymbol{y}) \geq \mathbf{0}} f(\boldsymbol{x}, \boldsymbol{y}) - \min_{\boldsymbol{x} \in \mathcal{X}} \max_{\boldsymbol{y} \in \mathcal{Y}: g(\boldsymbol{x}, \boldsymbol{y}) \geq \mathbf{0}} h(\boldsymbol{x}, \boldsymbol{y}) \right| \leq \max_{(\boldsymbol{x}, \boldsymbol{y}) \in \mathcal{X} \times \mathcal{Y}} |f(\boldsymbol{x}, \boldsymbol{y}) - h(\boldsymbol{x}, \boldsymbol{y})|$.*

With the above lemma in hand, we can now prove that $C$ is a contraction mapping.

**Theorem 2.3.** *Consider the operator $C$ associated with a stochastic Stackelberg game $\mathcal{G}^{(0)}$. Under Assumption 1.1, $C$ is a contraction mapping w.r.t. to the sup norm $\|.\|_\infty$ with constant $\gamma$.*

---

[4] We note SE should technically be defined in terms of non-stationary policies; however, as we will show, stationary policies suffice, since SE exist in stationary policies.

[5] All omitted results and proofs can be found in the appendix.

*Proof of Theorem 2.3.* We will show that $C$ is a contraction mapping, which then by Banach fixed point theorem establish the result. Let $v, v' \in \mathcal{V}$ be any two state value functions and $q, q' \in \mathcal{Q}$ be the respective associated action-value functions. We then have by Lemma 2.2:

$$\|Cv - Cv'\|_\infty \leq \max_{s \in \mathcal{S}} \left| \min_{x \in \mathcal{X}} \max_{y \in \mathcal{Y}: g(s,x,y) \geq 0} q(s, x, y) - \min_{x \in \mathcal{X}} \max_{y \in \mathcal{Y}: g(s,x,y) \geq 0} q'(s, x, y) \right| \quad (3)$$

$$\leq \max_{s \in \mathcal{S}} \max_{(x,y) \in \mathcal{X} \times \mathcal{Y}} |q(s, x, y) - q'(s, x, y)| \quad (4)$$

Replacing the definition of the state-action value function in the above, we get that $C$ is a contraction mapping since $\gamma \in (0, 1)$:

$$\leq \max_{s \in \mathcal{S}} \max_{(x,y) \in \mathcal{X} \times \mathcal{Y}} \left| \mathbb{E}_{S' \sim p(\cdot|s,x,y)} [r(s, x, y) + \gamma v(S')] - \mathbb{E}_{S' \sim p(\cdot|s,x,y)} [r(s, x, y) + \gamma v'(S')] \right| \quad (5)$$

$$\leq \gamma \max_{s \in \mathcal{S}} \max_{(x,y) \in \mathcal{X} \times \mathcal{Y}} \left| \mathbb{E}_{S' \sim p(\cdot|s,x,y)} [v(S') - v'(S')] \right| \quad (6)$$

$$\leq \gamma \max_{s \in \mathcal{S}} \max_{(x,y) \in \mathcal{X} \times \mathcal{Y}} |v(s) - v'(s)| = \gamma \|v - v'\|_\infty \quad (7)$$

$\square$

Given an initial state-value function $v^{(0)} \in \mathcal{V}$, we define the **value iteration** process as $v^{(t+1)} = Cv^{(t)}$, for all $t \in \mathbb{N}_+$ (Algorithm 2). One way to interpret $v^{(t)}$ is as the function that returns the value $v^{(t)}(s)$ of each state $s \in \mathcal{S}$ in the $t$-stage zero-sum stochastic Stackelberg game starting at the last stage $t$ and continuing until stage 0, with terminal payoffs given by $v^{(0)}$. The following theorem, which is a consequence of Theorems 2.1 and 2.3, and the Banach fixed point theorem [57], not only proves the existence of a recSE, but further provides us with a means of computing a recSE via value iteration.

**Theorem 2.4.** *Consider a zero-sum stochastic Stackelberg game $\mathcal{G}^{(0)}$. Under Assumption 1.1, $\mathcal{G}^{(0)}$ has a unique value function $v^{\pi_x^* \pi_y^*}$ associated with all recSE $(\pi_x^*, \pi_y^*)$, which can be computed by iteratively applying $C$ to any initial state-value function $v^{(0)} \in \mathcal{V}$: i.e., $\lim_{t \to \infty} v^{(t)} = v^{\pi_x^* \pi_y^*}$.*

**Remark 2.5.** *Unlike Shapley's existence theorem for recursive Nash equilibria in zero-sum stochastic games, Theorem 2.4 does not require the payoff function to be convex-concave. The only conditions needed are continuity of the payoffs and constraints, and bounded payoffs. This makes the recSE a potentially useful solution concept, even for non-convex-non-concave stochastic games.*

Since a recSE is guaranteed to exist, and is by definition independent of the initial state distribution, we can infer that the recSE of any zero-sum stochastic Stackelberg game $\mathcal{G}^{(0)} = (\mathcal{S}, \mathcal{X}, \mathcal{Y}, \mu^{(0)}, r, g, p, \gamma)$ is independent of the initial state distribution $\mu^{(0)}$. Hence, in the remainder of the paper, we denote zero-sum stochastic Stackelberg games by $\mathcal{G} \doteq (\mathcal{S}, \mathcal{X}, \mathcal{Y}, r, g, p, \gamma)$.

Theorem 2.4 tells us that value iteration converges to the value function associated with a recSE. Additionally, under Assumption 1.1, recSE is computable in (weakly) polynomial time.[6]

**Theorem 2.6** (Convergence of Value Iteration). *Suppose value iteration is run on input $\mathcal{G}$. Let $(\pi_x^*, \pi_y^*)$ be recSE of $\mathcal{G}$ with value function $v^{\pi_x^* \pi_y^*}$. Under Assumption 1.1, if we initialize $v^{(0)}(s) = 0$, for all $s \in \mathcal{S}$, then for $k \geq \frac{1}{1-\gamma} \log \frac{r_{max}}{\epsilon(1-\gamma)}$, it holds that $v^{(k)}(s) - v^{\pi_x^* \pi_y^*}(s) \leq \epsilon$.*

## 3 Subdifferential Envelope Theorems and Optimality Conditions for Recursive Stackelberg Equilibrium

In this section, we derive optimality conditions for recursive Stackelberg equilibria. In particular, we provide necessary conditions for a policy profile to be a recSE of any zero-sum stochastic Stackelberg game, and show that under additional convexity assumptions, these conditions are also sufficient.

---

[6]This convergence is only weakly polynomial time, because the computation of the generalized min-max operator applied to an arbitrary continuous function is an NP-hard problem; it is at least as hard as non-convex optimization. If, however, we restrict attention to convex-concave stochastic Stackelberg games, then Stackelberg equilibria are computable in polynomial time.

The Benveniste-Scheinkman theorem characterizes the derivative of the optimal value function associated with a recursive optimization problem w.r.t. its parameters, when it is differentiable [58]. Our proofs of the necessary and sufficient optimality conditions rely on a novel subdifferential generalization (Theorem C.2, Appendix C) of this theorem, which applies even when the optimal value function is not differentiable. A consequence of our subdifferential version of the Benveniste-Scheinkman theorem is that we can easily derive the first-order necessary conditions for a policy profile to be a recSE of any zero-sum stochastic Stackelberg game $\mathcal{G}$ satisfying Assumption 1.1, under standard regularity conditions.

**Theorem 3.1.** *Consider a zero-sum stochastic Stackelberg game $\mathcal{G}$, where $\mathcal{X} = \{\boldsymbol{x} \in \mathbb{R}^n \mid q_1(\boldsymbol{x}) \leq 0, \ldots, q_p(\boldsymbol{x}) \leq 0\}$ and $\mathcal{Y} = \{\boldsymbol{y} \in \mathbb{R}^m \mid r_1(\boldsymbol{y}) \geq 0, \ldots, r_l(\boldsymbol{y}) \geq 0\}$ are convex. Let $\mathcal{L}_{\boldsymbol{s},\boldsymbol{x}}(\boldsymbol{y}, \boldsymbol{\lambda}) = r(\boldsymbol{s}, \boldsymbol{x}, \boldsymbol{y}) + \gamma \mathbb{E}_{\boldsymbol{S}' \sim p(\cdot|\boldsymbol{s},\boldsymbol{x},\boldsymbol{y})}[v(\boldsymbol{S}', \boldsymbol{x})] + \sum_{k=1}^d \lambda_k g_k(\boldsymbol{s}, \boldsymbol{x}, \boldsymbol{y})$ where $Cv = v$.*

*Suppose that Assumption 1.1 holds, and that 1. for all $\boldsymbol{s} \in \mathcal{S}$, $\max_{\boldsymbol{y} \in \mathcal{Y}: \boldsymbol{g}(\boldsymbol{s},\boldsymbol{x},\boldsymbol{y}) \geq 0} \{r(\boldsymbol{s}, \boldsymbol{x}, \boldsymbol{y}) + \gamma \mathbb{E}_{\boldsymbol{S}' \sim p(\cdot|\boldsymbol{s},\boldsymbol{x},\boldsymbol{y})}[v(\boldsymbol{S}', \boldsymbol{x})]\}$ is concave in $\boldsymbol{x}$, 2. $\nabla_{\boldsymbol{x}} r(\boldsymbol{s}, \boldsymbol{x}, \boldsymbol{y}), \nabla_{\boldsymbol{x}} g_1(\boldsymbol{s}, \boldsymbol{x}, \boldsymbol{y}), \ldots, \nabla_{\boldsymbol{x}} g_d(\boldsymbol{s}, \boldsymbol{x}, \boldsymbol{y}), \nabla_{\boldsymbol{y}} r(\boldsymbol{s}, \boldsymbol{x}, \boldsymbol{y}), \nabla_{\boldsymbol{y}} g_1(\boldsymbol{s}, \boldsymbol{x}, \boldsymbol{y}), \ldots, \nabla_{\boldsymbol{y}} g_d(\boldsymbol{s}, \boldsymbol{x}, \boldsymbol{y})$ exist, for all $\boldsymbol{s} \in \mathcal{S}, \boldsymbol{x} \in \mathcal{X}, \boldsymbol{y} \in \mathcal{Y}$, 4. $p(\boldsymbol{s}' \mid \boldsymbol{s}, \boldsymbol{x}, \boldsymbol{y})$ is continuous and differentiable in $(\boldsymbol{x}, \boldsymbol{y})$, and 5. Slater's condition holds, i.e., $\forall \boldsymbol{s} \in \mathcal{S}, \boldsymbol{x} \in \mathcal{X}, \exists \widehat{\boldsymbol{y}} \in \mathcal{Y}$ s.t. $g_k(\boldsymbol{s}, \boldsymbol{x}, \widehat{\boldsymbol{y}}) > 0$ for all $k = 1, \ldots, d$ and $r_j(\widehat{\boldsymbol{y}}) > 0$, for all $j = 1, \ldots, l$, and $\exists \boldsymbol{x} \in \mathbb{R}^n$ s.t. $q_k(\boldsymbol{x}) < 0$ for all $k = 1 \ldots, p$. Then, there exists $\boldsymbol{\mu}^* : \mathcal{S} \to \mathbb{R}_+^p$, $\boldsymbol{\lambda}^* : \mathcal{S} \times \mathcal{X} \to \mathbb{R}_+^d$, and $\boldsymbol{\nu}^* : \mathcal{S} \times \mathcal{X} \to \mathbb{R}_+^l$ s.t. a policy profile $(\boldsymbol{\pi}_{\boldsymbol{x}}^*, \boldsymbol{\pi}_{\boldsymbol{y}}^*) \in \mathcal{X}^{\mathcal{S}} \times \mathcal{Y}^{\mathcal{S}}$ is a recSE of $\mathcal{G}$ only if it satisfies the following conditions, for all $\boldsymbol{s} \in \mathcal{S}$:*

$$\nabla_{\boldsymbol{x}} \mathcal{L}_{\boldsymbol{s},\boldsymbol{\pi}_{\boldsymbol{x}}^*(\boldsymbol{s})}(\boldsymbol{\pi}_{\boldsymbol{y}}^*(\boldsymbol{s}), \boldsymbol{\lambda}^*(\boldsymbol{s}, \boldsymbol{\pi}_{\boldsymbol{x}}^*(\boldsymbol{s}))) + \sum_{k=1}^p \mu_k^*(\boldsymbol{s}) \nabla_{\boldsymbol{x}} q_k(\boldsymbol{\pi}_{\boldsymbol{x}}^*(\boldsymbol{s})) = 0 \tag{8}$$

$$\nabla_{\boldsymbol{y}} \mathcal{L}_{\boldsymbol{s},\boldsymbol{\pi}_{\boldsymbol{x}}^*(\boldsymbol{s})}(\boldsymbol{\pi}_{\boldsymbol{y}}^*(\boldsymbol{s}), \boldsymbol{\lambda}^*(\boldsymbol{s}, \boldsymbol{\pi}_{\boldsymbol{x}}^*(\boldsymbol{s}))) + \sum_{k=1}^l \nu_k^*(\boldsymbol{s}, \boldsymbol{\pi}_{\boldsymbol{x}}^*(\boldsymbol{s})) \nabla_{\boldsymbol{x}} r_k(\boldsymbol{\pi}_{\boldsymbol{y}}^*(\boldsymbol{s})) = 0 \tag{9}$$

$$\mu_k^*(\boldsymbol{s}) q_k(\boldsymbol{\pi}_{\boldsymbol{x}}^*(\boldsymbol{s})) = 0 \qquad\qquad q_k(\boldsymbol{\pi}_{\boldsymbol{x}}^*(\boldsymbol{s})) \leq 0 \quad \forall k \in [p] \tag{10}$$

$$g_k(\boldsymbol{s}, \boldsymbol{\pi}_{\boldsymbol{x}}^*(\boldsymbol{s}), \boldsymbol{\pi}_{\boldsymbol{y}}^*(\boldsymbol{s})) \geq 0 \qquad \lambda_k^*(\boldsymbol{s}, \boldsymbol{\pi}_{\boldsymbol{x}}^*(\boldsymbol{s})) g_k(\boldsymbol{s}, \boldsymbol{\pi}_{\boldsymbol{x}}^*(\boldsymbol{s}), \boldsymbol{\pi}_{\boldsymbol{y}}^*(\boldsymbol{s})) = 0 \quad \forall k \in [d] \tag{11}$$

$$\nu_k^*(\boldsymbol{s}, \boldsymbol{\pi}_{\boldsymbol{x}}^*(\boldsymbol{s})) \nabla_{\boldsymbol{x}} r_k(\boldsymbol{\pi}_{\boldsymbol{y}}^*(\boldsymbol{s})) = 0 \qquad\qquad r_k(\boldsymbol{\pi}_{\boldsymbol{x}}^*(\boldsymbol{s})) \geq 0 \quad \forall k \in [l] \tag{12}$$

Under the conditions of Theorem 3.1, if we additionally assume that for all $\boldsymbol{s} \in \mathcal{S}$ and $\boldsymbol{x} \in \mathcal{X}$, both $r(\boldsymbol{s}, \boldsymbol{x}, \boldsymbol{y})$ and $g_1(\boldsymbol{s}, \boldsymbol{x}, \boldsymbol{y}), \ldots, g_d(\boldsymbol{s}, \boldsymbol{x}, \boldsymbol{y})$ are concave in $\boldsymbol{y}$, and $p(\boldsymbol{s}' \mid \boldsymbol{s}, \boldsymbol{x}, \boldsymbol{y})$ is continuous, CSD concave in $\boldsymbol{y}$, and differentiable in $(\boldsymbol{x}, \boldsymbol{y})$, Equations (59) to (63) become necessary *and sufficient* optimality conditions. For completeness, the reader can find the necessary and sufficient optimality conditions for convex-concave stochastic Stackelberg games under standard regularity conditions in Theorem C.3 (Appendix C). The proof follows exactly as that of Theorem 2.1.

## 4 Recursive Market Equilibrium

We now introduce an application of zero-sum stochastic Stackelberg games, which generalizes a well known market model, the Fisher market [12], to a dynamic setting in which buyers not only participate in markets across time, but their wealth persists. A **(static) Fisher market** consists of $n$ buyers and $m$ divisible goods [12]. Each buyer $i \in [n]$ is endowed with a budget $b_i \in \mathcal{B}_i \subseteq \mathbb{R}_+$ and a utility function $u_i : \mathbb{R}_+^m \times \mathcal{T}_i \to \mathbb{R}$, which is parameterized by a type $\boldsymbol{t}_i \in \mathcal{T}_i$ that defines a preference relation over the consumption space $\mathbb{R}_+^m$. Each good is characterized by a supply $q_j \in \mathcal{Q}_j \subset \mathbb{R}_+$.

An instance of a Fisher market is then a tuple $\mathcal{M} \doteq (n, m, \mathcal{U}, \boldsymbol{T}, \boldsymbol{b}, \boldsymbol{q})$, where $\mathcal{U} = \{u_1, \ldots, u_n\}$ is a set of utility functions, one per buyer, $\boldsymbol{b} \in \mathbb{R}_+^n$ is the vector of buyer budgets, and $\boldsymbol{q} \in \mathbb{R}_+^m$ is the vector of supplies. When clear from context, we simply denote $\mathcal{M}$ by $(\boldsymbol{T}, \boldsymbol{b}, \boldsymbol{q})$.

A **stochastic Fisher market with savings** is a dynamic market in which each state corresponds to a static Fisher market: i.e., each state $\boldsymbol{s} \in \mathcal{S}$ is characterized by a tuple $\boldsymbol{s} \doteq (\boldsymbol{T}, \boldsymbol{b}, \boldsymbol{q})$. At each state, the market determines the prices $\boldsymbol{p}$ of the goods, while the buyers choose their allocations $\boldsymbol{X} = (\boldsymbol{x}_1, \ldots, \boldsymbol{x}_n)^T \in \mathbb{R}_+^{n \times m}$ and potentially set aside some **savings** $\beta_i \in [0, b_i]$ to spend at some future state. Once allocations, savings, and prices have been determined, the market terminates with

probability $1 - \gamma$, or it transitions to a new state $\boldsymbol{s}'$ with probability $\gamma p(\boldsymbol{s}' \mid \boldsymbol{s}, \boldsymbol{\beta})$, depending on the buyers' saving decisions.[7] We denote a stochastic Fisher market by $\mathcal{F}^{(0)} \doteq (n, m, \mathcal{U}, \mathcal{S}, \boldsymbol{s}^{(0)}, p, \gamma)$.

Given a stochastic Fisher market with savings $\mathcal{F}^{(0)}$ a **recursive competitive equilibrium (recCE)** [15] is a tuple $(\boldsymbol{X}^*, \boldsymbol{\beta}^*, \boldsymbol{p}^*) \in \mathbb{R}_+^{n \times m \times \mathcal{S}} \times \mathbb{R}_+^{n \times \mathcal{S}} \times \mathbb{R}_+^{m \times \mathcal{S}}$, which consists of an allocation, savings, and price system s.t. 1) the buyers are expected utility maximizing, constrained by their savings and spending constraints, i.e., for all buyers $i \in [n]$, $(\boldsymbol{x}_i^*, \beta_i^*)$ is an optimal policy that, for all states $\boldsymbol{s} \doteq (\boldsymbol{T}, \boldsymbol{b}, \boldsymbol{q}) \in \mathcal{S}$, solves the **consumption-savings problem**, defined by the following Bellman equations: for all $\boldsymbol{s} \in \mathcal{S}$, $\nu_i(\boldsymbol{s}) =$

$$
\max_{(\boldsymbol{x}_i, \beta_i) \in \mathbb{R}_+^{m+1} : \boldsymbol{x}_i \cdot \boldsymbol{p}^*(\boldsymbol{s}) + \beta_i \leq b_i} \left\{ u_i(\boldsymbol{x}_i, \boldsymbol{t}_i) + \gamma \mathop{\mathbb{E}}_{(\boldsymbol{T}', \boldsymbol{b}', \boldsymbol{q}') \sim p(\cdot \mid \boldsymbol{s}, (\boldsymbol{x}_i, \boldsymbol{X}_{-i}^*(\boldsymbol{s})), (\beta_i, \boldsymbol{\beta}_{-i}^*(\boldsymbol{s})))} [\nu_i(\boldsymbol{T}', \boldsymbol{b}' + \beta_i, \boldsymbol{q}')] \right\},
$$

where $\boldsymbol{X}_{-i}^*, \boldsymbol{\beta}_{-i}^*$ denote the allocation and saving systems excluding buyer $i$; and 2) the market clears in each state so that unallocated goods in each state are priced at 0, i.e., for all $j \in [m]$ and $\boldsymbol{s} \in \mathcal{S}$, $p_j^*(\boldsymbol{s}) > 0 \implies \sum_{i \in [n]} x_{ij}^*(\boldsymbol{s}) = q_j$ and $p_j^*(\boldsymbol{s}) \geq 0 \implies \sum_{i \in [n]} x_{ij}^*(\boldsymbol{s}) \leq q_j$. By analogy with subgame perfect equilibrium, we can view a recCE as a "submarket" perfect equilibrium, as a recCE corresponds to a *competitive equilibrium* of the market starting from any state, i.e., buyers are allocated expected discounted cumulative utility-maximizing goods starting at any state, and the aggregate demand for any good is equal to its aggregate supply at all encountered markets.

The following theorem states that any recSE of a stochastic Fisher market with savings is in fact a recCE. Since recSE are guaranteed to exist, recCE are also guaranteed to exist. As recSE, and hence recCE, are independent of the initial market state, we denote any stochastic Fisher market with savings by $\mathcal{F} \doteq (n, m, \mathcal{U}, \mathcal{S}, p, \gamma)$.

**Theorem 4.1.** *A stochastic Fisher market with savings $\mathcal{F}$ in which $\mathcal{U}$ is a set of continuous and homogeneous utility functions and the transition function is continuous in $\beta_i$ has at least one recCE. Additionally, the recSE that solves the following Bellman equation corresponds to a recCE of $\mathcal{F}$:*

$$
v(\boldsymbol{s}) = \min_{\boldsymbol{p} \in \mathbb{R}_+^m} \max_{(\boldsymbol{X}, \boldsymbol{\beta}) \in \mathbb{R}_+^{n \times (m+1)} : \boldsymbol{X}\boldsymbol{p} + \boldsymbol{\beta} \leq \boldsymbol{b}} \sum_{j \in [m]} q_j p_j + \sum_{i \in [n]} (b_i - \beta_i) \log(u_i(\boldsymbol{x}_i, \boldsymbol{t}_i))
$$
$$
+ \gamma \mathop{\mathbb{E}}_{(\boldsymbol{T}', \boldsymbol{b}', \boldsymbol{q}') \sim p(\cdot \mid \boldsymbol{s}, \boldsymbol{\beta})} [v(\boldsymbol{T}', \boldsymbol{b}' + \boldsymbol{\beta}, \boldsymbol{q}')] \qquad (13)
$$

**Remark 4.2.** *This result cannot be obtained by modifying the Lagrangian formulation, i.e., the simultaneous-move game form, of the Eisenberg-Gale program, because the inner maximization problem is convex-non-concave, while recursive Nash equilibria are guaranteed to exist in zero-sum stochastic games only under the assumption of convex-concave payoffs [5].*

## 5 Experiments

The zero-sum stochastic Stackelberg game associated with a stochastic Fisher market, can, in theory, be solved via value iteration (Algorithm 1) assuming that one can compute the solution to the min-max optimization in line 4 of Algorithm 1. However, the min-max optimization problem which has to be solved at each step of value iteration is convex-non-concave. Specifically, the $\sum_{i \in [n]} (b_i - \beta_i) \log(u_i(\boldsymbol{x}_i, \boldsymbol{t}_i))$ term renders the objective function, convex-non-concave. This means that in general one is not guaranteed compute a globally optimal solution to the min-max optimization problem in line 4 of Algorithm 1 and instead can only converge to a local min-max solution with known (first-order) methods, e.g. nested gradient descent ascent [6]. Unfortunately, if we are not able to compute a globally optimal solution to the generalized min-max optimization, our guarantees for the convergence of value iteration do not apply. That said, gradient methods have been observed to escape local solutions in many non-convex optimization problems (e.g., see [60, 61]) leading us to investigate how well we can solve the generalized min-max operator in Algorithm 1 using nested gradient descent ascent [6], and in turn how effectively we can implement value iteration (Algorithm 1) in practice.

---

[7]In our model, which is consistent with the literature [59] 1. prices do not determine the next state since market prices are set by a "fictional auctioneer," not an actual market participant; 2. allocations do not determine the next state. Only savings, which are forward-looking decisions, affect future states—budgets, specifically.

---

**Algorithm 1** Value Iteration for Stochastic Fisher Market

---

1: Initialize $v^{(0)}$ arbitrarily, e.g. $v^{(0)} = \mathbf{0}$
2: **for** $k = 1, \ldots, T_v$ **do**
3:     For all $\boldsymbol{s} \in \mathcal{S}$, $v^{(k+1)}(\boldsymbol{T}, \boldsymbol{b}, \boldsymbol{q}) =$
4: $\min_{\boldsymbol{p} \geq \boldsymbol{0}} \max_{(\boldsymbol{X}, \boldsymbol{\beta}) \geq \boldsymbol{0}: \boldsymbol{Xp} + \boldsymbol{\beta} \leq \boldsymbol{b}} \left\{ \sum_{j \in [m]} q_j p_j + \sum_{i \in [n]} (b_i - \beta_i) \log(u_i(\boldsymbol{x}_i, \boldsymbol{t}_i)) + \gamma \, \mathbb{E} \left[ v^{(k)}(\boldsymbol{T}', \boldsymbol{b}' + \boldsymbol{\beta}, \boldsymbol{q}') \right] \right\}$

---

To do so, we computed the recursive Stackelberg equilibria of three different classes of stochastic Fisher markets with savings.[8] Specifically, we created markets with three classes of utility functions, each of which endowed the state-value function with different smoothness properties. Let $\boldsymbol{t}_i \in \mathbb{R}^m$ be a vector of parameters, i.e., a **type**, that describes the utility function of buyer $i \in [n]$. We considered the following (standard) utility function classes: 1. **linear**: $u_i(\boldsymbol{x}_i) = \sum_{j \in [m]} t_{ij} x_{ij}$;

2. **Cobb-Douglas**: $u_i(\boldsymbol{x}_i) = \prod_{j \in [m]} x_{ij}^{t_{ij}}$; and 3. **Leontief**: $u_i(\boldsymbol{x}_i) = \min_{j \in [m]} \left\{ \frac{x_{ij}}{t_{ij}} \right\}$.

We ran two different experiments. First, we modeled a small stochastic Fisher market with savings *without* interest rates. In this setting, buyers' budgets, which are initialized at the start of the game, persist across states, and are replenished by a constant amount with each state transition. Thus, the buyers' budgets from one state to the next are deterministic.

Second, we modeled a larger stochastic Fisher market with savings and probabilistic interest rates. In this model, although buyers' savings persist across states, they are nondeterministic, as they increase or decrease based on the random movements of an interest rate with each state transition. More specifically, we chose five different equiprobable interest rates (0.9, 1.0, 1.1, 1.2, and 1.5) to provide buyers with more incentive to save as compared to the model without interest rates.

Since budgets are a part of the state space in stochastic Fisher markets, the state space is continuous; so we attempted to estimate the value function at each iteration of value iteration by running linear regression on a sample of state and associated min-max value pairs, finding a fit via linear regression (e.g., [62]). To compute the min-max value of each state that we sampled, i.e., the solution to the optimization problem in line 4 of Algorithm 1, we used nested gradient descent ascent [6] which runs a step of gradient descent on the prices and a loop of gradient ascent on allocations and savings repeatedly (Algorithm 3), where we computed gradients via auto-differentiation using JAX [63] which we observed achieved better numerical stability than analytically derived gradients as can often be the case with autodifferentiation [64].

In both experiments, to check whether the optimal value function was found, we measured the exploitability of the market, meaning the distance between the recCE computed and the actual recCE. To do so, we checked two conditions: 1) whether each buyer's expected utility was maximized at the computed allocation and savings, at the prices outputted by the algorithm, and 2) whether the market always cleared. In both settings, we extracted the greedy policy from the value function computed by value iteration, and unrolled it across time to obtain the greedy actions $(\boldsymbol{X}^{(t)}, \boldsymbol{\beta}^{(t)}, \boldsymbol{p}^{(t)})$ at each state $\boldsymbol{s}^{(t)}$. We then computed the cumulative utility of these allocation and savings, i.e., for all $i \in [n]$, $\hat{u}_i \doteq \sum_{t=0}^{T} \gamma^t u_i(\boldsymbol{x}_i^{(t)})$. We compared these values to the expected maximum utility $u_i^*$, given the prices and the other buyers' allocations computed by our algorithm. We report the normalized distance between these two values, $\hat{u}_i$ and $u_i^*$, which we call the **normalized distance to utility maximization (UM)**. For example, in the case of two buyers, the normalized distance to UM $= \frac{||(\hat{u}_1, \hat{u}_2) - (u_1^*, u_2^*)||_2}{||(u_1^*, u_2^*)||_2}$. Finally, we also measured excess demand, which we took as the **distance to market clearance (MC)**, i.e., $\frac{1}{T} \sum_{t=1}^{T} ||\sum_{i \in [n]} \boldsymbol{x}_i^{(t)} - \boldsymbol{q}^{(t)}||_2$.

In the experiment with smaller markets and without interest rates, Figure 1 depicts the average value of the value function across a sample of states as it varies with time, and Table 1 records the exploitability of the recCE found by nested GDA. For all three class of utility functions, not only do the value functions converge, exploitability is also sufficiently minimized, as all the buyer utilities are maximized and the market always clears.

---

[8]Our code can be found here, and details of our experimental setup can be found in Appendix E.

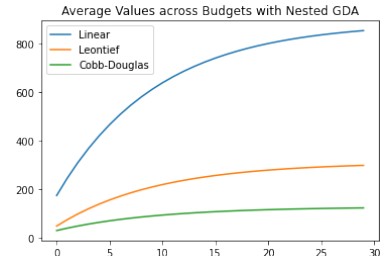

Figure 1: The value function averaged across budgets.

| Utility Class | Distance to UM | Distance to MC |
|---|---|---|
| Linear | 0.011 | 0.010 |
| Leontief | 0.056 | 0.010 |
| Cobb-Douglas | 0.006 | 0.010 |

Table 1: Exploitability of recCE found by Nested GDA.

Table 2: Nested GDA in stochastic Fisher markets with savings but without interest rates.

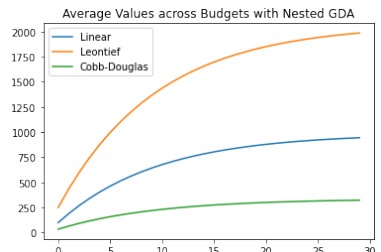

Figure 2: The value function averaged across budgets.

| Utility Class | Distance to UM | Distance to MC |
|---|---|---|
| Linear | 0.040 | 0.009 |
| Leontief | 0.463 | 0.009 |
| Cobb-Douglas | 0.017 | 0.009 |

Table 3: Exploitability of recCE found by Nested GDA.

Table 4: Nested GDA in stochastic Fisher markets with savings and probabilistic interest rates.

In the experiment with larger markets and probabilistic interest rates (Figure 2, Table 3), in linear and Cobb-Douglas markets, the value functions converge, and exploitability is sufficiently minimized. In Leontief markets, however, although the value function converges and the markets almost clear, the buyers' utilities are not fully maximized, since the cumulative utilities they obtained are less than half of their expected maximum utilities. The difficulty in this case likely arises from the fact that the Leontief utility function is not differentiable, so the problem for Leontief markets is neither smooth nor convex-concave, which makes it difficult, if not impossible, for nested GDA to find even a stationary point of the objective of the min-max optimization problem in line 4 of Algorithm 1, since gradient ascent on a function is not guaranteed to converge to a stationary point of that function if it is non-convex-non-smooth [65].

## 6  Conclusion

In this paper, we proved the existence of recursive Stackelberg equilibria in zero-sum stochastic Stackelberg games, provided necessary and sufficient conditions for a policy profile to be a recursive Stackelberg equilibrium, and showed that a Stackelberg equilibrium can be computed in (weakly) polynomial time via value iteration. Finally, we showed that recursive Stackelberg equilibria coincide with recursive competitive equilibria in stochastic Fisher markets, and we used value iteration together with nested GDA to solve for them. Future work in this space could try using deep reinforcement learning methods to learn better (i.e., nonlinear) representations of the value functions. It is also conceivable that deep reinforcement learning would be able to learn better policies, thereby resolving the difficulties that our methods face due to non-smoothness and non-concavity have in solving for global solutions of the min-max optimization problem in line 4 of Algorithm 1 .

## Acknowledgments and Disclosure of Funding

This work was supported by NSF Grant CMMI-1761546.

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
