# A  Pseudo-Code for Algorithms

---

**Algorithm 2** Value Iteration (with Min-Max Oracle)

---

**Inputs:** $\mathcal{S}, \mathcal{X}, \mathcal{Y}, r, \boldsymbol{g}, p, \gamma, T$
**Outputs:** $v^{(T)}$

1: Initialize $v^{(0)}$ arbitrarily, e.g. $v^{(0)} = 0$
2: **for** $t = 1, \ldots, T$ **do**
3:     **for** $\boldsymbol{s} \in \mathcal{S}$ **do**
4:         $v^{(t+1)}(\boldsymbol{s}) = \min\limits_{\boldsymbol{x} \in \mathcal{X}} \max\limits_{\boldsymbol{y} \in \mathcal{Y}: \boldsymbol{g}(\boldsymbol{s}, \boldsymbol{x}, \boldsymbol{y}) \geq \boldsymbol{0}} \mathbb{E}\limits_{\boldsymbol{S}' \sim p(\cdot|\boldsymbol{s}, \boldsymbol{x}, \boldsymbol{y})} \left[ r(\boldsymbol{s}, \boldsymbol{x}, \boldsymbol{y}) + \gamma v^{(t)}(\boldsymbol{S}') \right]$
5: **return** $v^{(T)}$

---

**Algorithm 3** Nested GDA for stochastic Fisher Markets with saving

---

**Inputs:** $v, \boldsymbol{T}, \boldsymbol{b}, \boldsymbol{q}, \eta_{\boldsymbol{p}}, \eta_{\boldsymbol{X}}, T_{\boldsymbol{p}}, T_{\boldsymbol{X}}, \boldsymbol{p}^{(0)}, \boldsymbol{X}^{(0)}, \boldsymbol{\beta}^{(0)}$
**Output:** $(\boldsymbol{p}^{(t)}, \boldsymbol{X}^{(t)})_{t=1}^{T}$

1: **for** $t = 1, \ldots, T_{\boldsymbol{p}}$ **do**
2:     **for** $s = 1, \ldots, T_{\boldsymbol{X}}$ **do**
3:         For all $i \in [n]$, $\boldsymbol{x}_i^{(t)} = \boldsymbol{x}_i^{(t)} + \eta_{\boldsymbol{X}} \left( \dfrac{b_i - \beta_i^{(t)}}{u_i\left(\boldsymbol{x}_i^{(t)}; \boldsymbol{t}_i\right)} \nabla_{\boldsymbol{x}_i} u_i\left(\boldsymbol{x}_i^{(t)}; \boldsymbol{t}_i\right) \right)$
4:         For all $i \in [n]$, $\beta_i^{(t)} = \beta_i^{(t)} + \eta_{\boldsymbol{X}} \left( -\log(u_i(\boldsymbol{x}_i^{(t)}; \boldsymbol{t}_i)) + \gamma \dfrac{\partial v(\boldsymbol{T}, \boldsymbol{b}, \boldsymbol{q})}{\partial b_i} \right)$
5:         $(\boldsymbol{X}^{(t)}, \boldsymbol{\beta}^{(t)}) = \Pi_{\{(\boldsymbol{X}, \boldsymbol{\beta}) \in \mathbb{R}_+^{n \times m} \times \mathbb{R}_+^n : \boldsymbol{X} \cdot \boldsymbol{p}^{(t-1)} + \boldsymbol{\beta} \leq \boldsymbol{b}\}} \left( (\boldsymbol{X}^{(t)}, \boldsymbol{\beta}^{(t)}) \right)$
6:     $\boldsymbol{p}^{(t)} = \Pi_{\mathbb{R}_+^m} \left( \boldsymbol{p}^{(t-1)} - \eta_{\boldsymbol{p}}(\boldsymbol{1} - \sum_{i \in [n]} \boldsymbol{x}_i^{(t)}) \right)$
7: **return** $(\boldsymbol{p}^{(t)}, \boldsymbol{X}^{(t)})_{t=1}^{T_{\boldsymbol{p}}}$

---

# B  Omitted Results and Proofs Section 2

We first note the following fundamental relationship between the state-value and action-value functions which is an analog of Bellman's Theorem [66] and which follows from their definitions:

**Theorem B.1.** *Given a stochastic min-max Stackelberg game* $(\mathcal{S}, \mathcal{X}, \mathcal{Y}, \mu^{(0)}, r, \boldsymbol{g}, p, \gamma)$, *for all* $v \in \mathcal{V}$, $q \in \mathcal{Q}$, $\boldsymbol{\pi}_{\boldsymbol{x}} \in \mathcal{X}^{\mathcal{S}}$, *and* $\boldsymbol{\pi}_{\boldsymbol{y}} \in \mathcal{Y}^{\mathcal{S}}$, $v = v^{\boldsymbol{\pi}_{\boldsymbol{x}} \boldsymbol{\pi}_{\boldsymbol{y}}}$ *and* $q = q^{\boldsymbol{\pi}_{\boldsymbol{x}} \boldsymbol{\pi}_{\boldsymbol{y}}}$ *iff:*

$$v(\boldsymbol{s}) = q(\boldsymbol{s}, \boldsymbol{\pi}_{\boldsymbol{x}}(\boldsymbol{s}), \boldsymbol{\pi}_{\boldsymbol{y}}(\boldsymbol{s})) \tag{14}$$

$$q(\boldsymbol{s}, \boldsymbol{x}, \boldsymbol{y}) = \mathbb{E}_{\boldsymbol{S}' \sim p(\cdot|\boldsymbol{s}, \boldsymbol{x}, \boldsymbol{y})} [r(\boldsymbol{s}, \boldsymbol{x}, \boldsymbol{y}) + \gamma v(\boldsymbol{S}')] \tag{15}$$

*Proof of Theorem B.1.* By the definition of the state value function we have $v_i^{\boldsymbol{\pi}_{\boldsymbol{x}} \boldsymbol{\pi}_{\boldsymbol{y}}} = q_i(\boldsymbol{s}, \boldsymbol{\pi}_{\boldsymbol{x}}(\boldsymbol{s}), \boldsymbol{\pi}_{\boldsymbol{y}}(\boldsymbol{s}))$, hence by Equation (14) we must have that $v_i = v_i^{\boldsymbol{\pi}_{\boldsymbol{x}} \boldsymbol{\pi}_{\boldsymbol{y}}}$. Additionally, by Equation (15) and the definition of the action-value functions this also implies that $q_i(\boldsymbol{s}, \boldsymbol{x}, \boldsymbol{y}) = q_i^{\boldsymbol{\pi}_{\boldsymbol{x}} \boldsymbol{\pi}_{\boldsymbol{y}}}(\boldsymbol{s}, \boldsymbol{x}, \boldsymbol{y})$ $\qquad \square$

**Theorem 2.1.** $(\boldsymbol{\pi}_{\boldsymbol{x}}^*, \boldsymbol{\pi}_{\boldsymbol{y}}^*)$ *is a recSE of* $\mathcal{G}^{(0)}$ *of* $v^{\boldsymbol{\pi}_{\boldsymbol{x}} \boldsymbol{\pi}_{\boldsymbol{y}}}$ *iff it induces a value function which is a fixed point of* $C$: *i.e.,* $(\boldsymbol{\pi}_{\boldsymbol{x}}^*, \boldsymbol{\pi}_{\boldsymbol{y}}^*)$ *is a Stackelberg equilbrium iff, for all* $\boldsymbol{s} \in \mathcal{S}$, $\left( C v^{\boldsymbol{\pi}_{\boldsymbol{x}}^* \boldsymbol{\pi}_{\boldsymbol{y}}^*} \right)(\boldsymbol{s}) = v^{\boldsymbol{\pi}_{\boldsymbol{x}}^* \boldsymbol{\pi}_{\boldsymbol{y}}^*}(\boldsymbol{s})$.

*Proof of Theorem 2.1.* We prove one direction, the other direction follows symmetrically. (Fixed Point $\implies$ recursive Stackelberg equilibrium) Suppose that a value function $v^{\boldsymbol{\pi}_{\boldsymbol{x}}^* \boldsymbol{\pi}_{\boldsymbol{y}}^*}$ which is induced

by a policy profile $(\boldsymbol{\pi}_{\boldsymbol{x}}^*, \boldsymbol{\pi}_{\boldsymbol{y}}^*)$ is a fixed point of $C$, we then have for all states $\boldsymbol{s} \in \mathcal{S}$:

$$v^{\boldsymbol{\pi}_{\boldsymbol{x}}^* \boldsymbol{\pi}_{\boldsymbol{y}}^*} = \left( C v^{\boldsymbol{\pi}_{\boldsymbol{x}}^* \boldsymbol{\pi}_{\boldsymbol{y}}^*} \right) (\boldsymbol{s}) \tag{16}$$

$$= \min_{\boldsymbol{x} \in \mathcal{X}} \max_{\boldsymbol{y} \in \mathcal{Y}: \boldsymbol{g}(\boldsymbol{s}, \boldsymbol{x}, \boldsymbol{y}) \geq \mathbf{0}} \mathbb{E}_{\boldsymbol{S}' \sim p(\cdot | \boldsymbol{s}, \boldsymbol{x}, \boldsymbol{y})} \left[ r(\boldsymbol{s}, \boldsymbol{x}, \boldsymbol{y}) + \gamma v^{\boldsymbol{\pi}_{\boldsymbol{x}}^* \boldsymbol{\pi}_{\boldsymbol{y}}^*}(\boldsymbol{S}') \right] \tag{17}$$

$$= \min_{\boldsymbol{x} \in \mathcal{X}} \max_{\boldsymbol{y} \in \mathcal{Y}: \boldsymbol{g}(\boldsymbol{s}, \boldsymbol{x}, \boldsymbol{y}) \geq \mathbf{0}} q^{\boldsymbol{\pi}_{\boldsymbol{x}}^* \boldsymbol{\pi}_{\boldsymbol{y}}^*}(\boldsymbol{s}, \boldsymbol{x}, \boldsymbol{y}) \tag{18}$$

Hence, by Definition 1.2, $(\boldsymbol{\pi}_{\boldsymbol{x}}^*, \boldsymbol{\pi}_{\boldsymbol{y}}^*)$ is recursive Stackelberg equilibrium. $\qquad\square$

**Lemma 2.2.** *Suppose that $f, h : \mathcal{X} \times \mathcal{Y} \to \mathbb{R}$, $\boldsymbol{g} : \mathcal{X} \times \mathcal{Y} \to \mathbb{R}^d$ are continuous functions, and $\mathcal{X}, \mathcal{Y}$ are compact sets. Then $\left| \min_{\boldsymbol{x} \in \mathcal{X}} \max_{\boldsymbol{y} \in \mathcal{Y}: \boldsymbol{g}(\boldsymbol{x}, \boldsymbol{y}) \geq \mathbf{0}} f(\boldsymbol{x}, \boldsymbol{y}) - \min_{\boldsymbol{x} \in \mathcal{X}} \max_{\boldsymbol{y} \in \mathcal{Y}: \boldsymbol{g}(\boldsymbol{x}, \boldsymbol{y}) \geq \mathbf{0}} h(\boldsymbol{x}, \boldsymbol{y}) \right| \leq \max_{(\boldsymbol{x}, \boldsymbol{y}) \in \mathcal{X} \times \mathcal{Y}} |f(\boldsymbol{x}, \boldsymbol{y}) - h(\boldsymbol{x}, \boldsymbol{y})|$.*

*Proof of Lemma 2.2.* Let $(\boldsymbol{x}^*, \boldsymbol{y}^*)$ be a Stackelberg equilibrium of $\min_{\boldsymbol{x} \in \mathcal{X}} \max_{\boldsymbol{y} \in \mathcal{Y}: \boldsymbol{g}(\boldsymbol{x}, \boldsymbol{y}) \geq \mathbf{0}} f(\boldsymbol{x}, \boldsymbol{y})$, and $(\boldsymbol{x}', \boldsymbol{y}')$ be a Stackelberg equilibrium of $\min_{\boldsymbol{x} \in \mathcal{X}} \max_{\boldsymbol{y} \in \mathcal{Y}: \boldsymbol{g}(\boldsymbol{x}, \boldsymbol{y}) \geq \mathbf{0}} h(\boldsymbol{x}, \boldsymbol{y})$. Additionally, let $\bar{y} \in \arg\max_{\boldsymbol{y} \in \mathcal{Y}: \boldsymbol{g}(\boldsymbol{x}', \boldsymbol{y}) \geq \mathbf{0}} f(\boldsymbol{x}', \boldsymbol{y})$, then by the the definition of a Stackelberg equilibrium, we have $f(\boldsymbol{x}^*, \boldsymbol{y}^*) = \min_{\boldsymbol{y} \in \mathcal{Y}} \max_{\boldsymbol{x} \in \mathcal{X}: \boldsymbol{g}(\boldsymbol{x}, \boldsymbol{y}) \geq \mathbf{0}} f(\boldsymbol{x}, \boldsymbol{y}) \leq \max_{\boldsymbol{y} \in \mathcal{Y}: \boldsymbol{g}(\boldsymbol{x}', \boldsymbol{y}) \geq \mathbf{0}} f(\boldsymbol{x}', \boldsymbol{y}) = f(\boldsymbol{x}', \bar{y})$, and $h(\boldsymbol{x}', \boldsymbol{y}') = \max_{\boldsymbol{y} \in \mathcal{Y}: \boldsymbol{g}(\boldsymbol{x}', \boldsymbol{y}) \geq \mathbf{0}} h(\boldsymbol{x}', \boldsymbol{y}) \geq h(\boldsymbol{x}', \bar{y})$.

Suppose that $\min_{\boldsymbol{x} \in \mathcal{X}} \max_{\boldsymbol{y} \in \mathcal{Y}: \boldsymbol{g}(\boldsymbol{x}, \boldsymbol{y}) \geq \mathbf{0}} f(\boldsymbol{x}, \boldsymbol{y}) \geq \min_{\boldsymbol{x} \in \mathcal{X}} \max_{\boldsymbol{y} \in \mathcal{Y}: \boldsymbol{g}(\boldsymbol{x}, \boldsymbol{y}) \geq \mathbf{0}}, h(\boldsymbol{x}, \boldsymbol{y})$ this gives us:

$$\left| \min_{\boldsymbol{x} \in \mathcal{X}} \max_{\boldsymbol{y} \in \mathcal{Y}: \boldsymbol{g}(\boldsymbol{x}, \boldsymbol{y}) \geq \mathbf{0}} f(\boldsymbol{x}, \boldsymbol{y}) - \min_{\boldsymbol{x} \in \mathcal{X}} \max_{\boldsymbol{y} \in \mathcal{Y}: \boldsymbol{g}(\boldsymbol{x}, \boldsymbol{y}) \geq \mathbf{0}} h(\boldsymbol{x}, \boldsymbol{y}) \right| \tag{19}$$

$$= |f(\boldsymbol{x}^*, \boldsymbol{y}^*) - h(\boldsymbol{x}', \boldsymbol{y}')| \tag{20}$$

$$\leq |f(\boldsymbol{x}', \bar{y}) - h(\boldsymbol{x}', \boldsymbol{y}')| \tag{21}$$

$$\leq |f(\boldsymbol{x}', \bar{y}) - h(\boldsymbol{x}', \bar{y})| \tag{22}$$

$$\leq \max_{(\boldsymbol{x}, \boldsymbol{y}) \in \mathcal{X} \times \mathcal{Y}} |f(\boldsymbol{x}, \boldsymbol{y}) - h(\boldsymbol{x}, \boldsymbol{y})| \tag{23}$$

The opposite case follows similarly by symmetry. $\qquad\square$

**Theorem 2.3.** *Consider the operator $C$ associated with a stochastic Stackelberg game $\mathcal{G}^{(0)}$. Under Assumption 1.1, $C$ is a contraction mapping w.r.t. to the sup norm $\|.\|_\infty$ with constant $\gamma$.*

*Proof of Theorem 2.3.* We will show that $C$ is a contraction mapping, which then by Banach fixed point theorem establish the result. Let $v, v' \in \mathcal{V}$ be any two state value functions and $q, q' \in \mathcal{Q}$ be the respective associated action-value functions. We then have:

$$\|Cv - Cv'\|_\infty \tag{24}$$

$$\leq \max_{\boldsymbol{s} \in \mathcal{S}} \left| \min_{\boldsymbol{x} \in \mathcal{X}} \max_{\boldsymbol{y} \in \mathcal{Y}: \boldsymbol{g}(\boldsymbol{s}, \boldsymbol{x}, \boldsymbol{y}) \geq \mathbf{0}} q(\boldsymbol{s}, \boldsymbol{x}, \boldsymbol{y}) - \min_{\boldsymbol{x} \in \mathcal{X}} \max_{\boldsymbol{y} \in \mathcal{Y}: \boldsymbol{g}(\boldsymbol{s}, \boldsymbol{x}, \boldsymbol{y}) \geq \mathbf{0}} q'(\boldsymbol{s}, \boldsymbol{x}, \boldsymbol{y}) \right| \tag{25}$$

$$\leq \max_{\boldsymbol{s} \in \mathcal{S}} \max_{(\boldsymbol{x}, \boldsymbol{y}) \in \mathcal{X} \times \mathcal{Y}} |q(\boldsymbol{s}, \boldsymbol{x}, \boldsymbol{y}) - q'(\boldsymbol{s}, \boldsymbol{x}, \boldsymbol{y})| \qquad\qquad \text{(Lemma 2.2)} \tag{26}$$

$$\leq \max_{\boldsymbol{s} \in \mathcal{S}} \max_{(\boldsymbol{x}, \boldsymbol{y}) \in \mathcal{X} \times \mathcal{Y}} \left| \mathbb{E}_{\boldsymbol{S}' \sim p(\cdot | \boldsymbol{s}, \boldsymbol{x}, \boldsymbol{y})} [r(\boldsymbol{s}, \boldsymbol{x}, \boldsymbol{y}) + \gamma v(\boldsymbol{S}')] - \mathbb{E}_{\boldsymbol{S}' \sim p(\cdot | \boldsymbol{s}, \boldsymbol{x}, \boldsymbol{y})} [r(\boldsymbol{s}, \boldsymbol{x}, \boldsymbol{y}) + \gamma v'(\boldsymbol{S}')] \right| \tag{27}$$

$$\leq \max_{\boldsymbol{s} \in \mathcal{S}} \max_{(\boldsymbol{x}, \boldsymbol{y}) \in \mathcal{X} \times \mathcal{Y}} \left| \mathbb{E}_{\boldsymbol{S}' \sim p(\cdot | \boldsymbol{s}, \boldsymbol{x}, \boldsymbol{y})} [\gamma v(\boldsymbol{S}') - \gamma v'(\boldsymbol{S}')] \right| \tag{28}$$

$$\leq \gamma \max_{\boldsymbol{s} \in \mathcal{S}} \max_{(\boldsymbol{x}, \boldsymbol{y}) \in \mathcal{X} \times \mathcal{Y}} \left| \mathbb{E}_{\boldsymbol{S}' \sim p(\cdot | \boldsymbol{s}, \boldsymbol{x}, \boldsymbol{y})} [v(\boldsymbol{S}') - v'(\boldsymbol{S}')] \right| \tag{29}$$

$$\leq \gamma \max_{\boldsymbol{s} \in \mathcal{S}} \max_{(\boldsymbol{x}, \boldsymbol{y}) \in \mathcal{X} \times \mathcal{Y}} |v(\boldsymbol{s}) - v'(\boldsymbol{s})| \tag{30}$$

$$= \gamma \|v - v'\|_\infty \tag{31}$$

Since $\gamma \in (0,1)$, $C$ is a contraction mapping.

$\square$

**Theorem 2.4.** *Consider a zero-sum stochastic Stackelberg game $\mathcal{G}^{(0)}$. Under Assumption 1.1, $\mathcal{G}^{(0)}$ has a unique value function $v^{\pi_x^* \pi_y^*}$ associated with all recSE $(\pi_x^*, \pi_y^*)$, which can be computed by iteratively applying $C$ to any initial state-value function $v^{(0)} \in \mathcal{V}$: i.e., $\lim_{t \to \infty} v^{(t)} = v^{\pi_x^* \pi_y^*}$.*

*Proof of Theorem 2.4.* By combining Theorem 2.3 and the Banach fixed point theorem [57], we obtain that a fixed point of $C$ exists. Hence, by Theorem 2.1, a recursive Stackelberg equilibrium of $(\mathcal{S}, \mathcal{X}, \mathcal{Y}, \mu^{(0)}, r, \boldsymbol{g}, p, \gamma)$ exists and the value function induced by all recursive Stackelberg equilibria is the same, i.e., the optimal value function is unique. Additionally, by the second part of the Banach fixed point theorem, we must then also have $\lim_{t \to \infty} v^{(t)} = v^{\pi_x^* \pi_y^*}$. $\square$

For any $q \in \mathcal{Q}$, we define a **greedy policy profile** with respect to $q$ as a policy profile $(\pi_x^q, \pi_y^q)$ such that $\pi_x^q \in \arg\min_{\boldsymbol{x} \in \mathcal{X}} \max_{\boldsymbol{y} \in \mathcal{Y}: \boldsymbol{g}(\boldsymbol{s}, \boldsymbol{x}, \boldsymbol{y}) \geq \boldsymbol{0}} q(\boldsymbol{s}, \boldsymbol{x}, \boldsymbol{y})$ and $\pi_y^q \in \arg\max_{\boldsymbol{y} \in \mathcal{Y}: \boldsymbol{g}(\boldsymbol{s}, \pi_x^q(\boldsymbol{x}), \boldsymbol{y}) \geq \boldsymbol{0}} q(\boldsymbol{s}, \pi_x^q(\boldsymbol{x}), \boldsymbol{y})$. The following lemma provides a progress bound for each iteration of value iteration which is expressed in terms of the value function associated with the greedy policy profile.

**Theorem 2.6** (Convergence of Value Iteration). *Suppose value iteration is run on input $\mathcal{G}$. Let $(\pi_x^*, \pi_y^*)$ be recSE of $\mathcal{G}$ with value function $v^{\pi_x^* \pi_y^*}$. Under Assumption 1.1, if we initialize $v^{(0)}(\boldsymbol{s}) = 0$, for all $\boldsymbol{s} \in \mathcal{S}$, then for $k \geq \frac{1}{1-\gamma} \log \frac{r_{\max}}{\epsilon(1-\gamma)}$, it holds that $v^{(k)}(\boldsymbol{s}) - v^{\pi_x^* \pi_y^*}(\boldsymbol{s}) \leq \epsilon$.*

*Proof of Theorem 2.6.* First note that by Assumption 1.1, we have that $\left\| v^{\pi_x^* \pi_y^*} \right\|_\infty \leq \frac{r_{\max}}{1-\gamma}$. Applying the operator $C$ repeatedly and using the fact that $v^{\pi_x^* \pi_y^*} = C v^{\pi_x^* \pi_y^*}$ from Theorem 2.1, we obtain

$$\| v^{(k)} - v^{\pi_x^* \pi_y^*} \|_\infty \tag{32}$$

$$= \| (C)^k v^{(0)} - (C)^k v^{\pi_x^* \pi_y^*} \|_\infty \tag{33}$$

$$\leq \gamma^k \| v^{(0)} - v^{\pi_x^* \pi_y^*} \|_\infty \tag{34}$$

$$= \gamma^k \| v^{\pi_x^* \pi_y^*} \|_\infty \tag{35}$$

$$\leq \gamma^k \frac{r_{\max}}{1-\gamma} \tag{36}$$

where Equation (35) was obtained as $v^{(0)} = 0$ Since $1 - x \leq e^{-x}$ for any $x \in \mathbb{R}$, we have

$$\gamma^k = (1 - (1-\gamma))^k \leq (e^{-(1-\gamma)})^k \leq e^{-(1-\gamma)k}$$

Thus, for any $\boldsymbol{s} \in \mathcal{S}$

$$v^{(k)}(\boldsymbol{s}) - v^{\pi_x^* \pi_y^*}(\boldsymbol{s}) \leq \| v^{(k)} - v^{\pi_x^* \pi_y^*} \|_\infty$$

$$= \gamma^k \frac{r_{\max}}{1-\gamma}$$

$$\leq e^{-(1-\gamma)k} \frac{r_{\max}}{1-\gamma}$$

Thus it suffices to solve for $k$ such that

$$e^{-(1-\gamma)k} \frac{r_{\max}}{1-\gamma} \leq \varepsilon \ .$$

which concludes the proof. $\square$

## C   Omitted Results and Proofs Section 3

Our characterization of the subdifferential of the value function associated with a Stackelberg equilibrium w.r.t. its parameters relies on a slightly generalized version of the subdifferential envelope

theorem (Theorem C.1, Appendix C) of Goktas and Greenwald [6], which characterizes the set of subdifferentials of parametrized constrained optimization problems, i.e., the set of subgradients w.r.t. $\boldsymbol{x}$ of $f^*(\boldsymbol{x}) = \max_{\boldsymbol{y} \in \mathcal{Y}: h(\boldsymbol{x},\boldsymbol{y}) \geq 0} f(\boldsymbol{x}, \boldsymbol{y})$. In particular, we note that Goktas and Greenwald's proof goes through even without assuming the concavity of $f(\boldsymbol{x},\boldsymbol{y}), h_1(\boldsymbol{x},\boldsymbol{y}), \ldots, h_d(\boldsymbol{x},\boldsymbol{y})$ in $\boldsymbol{y}$, for all $\boldsymbol{x} \in \mathcal{X}$.

**Theorem C.1** (Subdifferential Envelope Theorem). *Consider the function $f^*(\boldsymbol{x}) = \max_{\boldsymbol{y} \in \mathcal{Y}: h(\boldsymbol{x},\boldsymbol{y}) \geq 0} f(\boldsymbol{x},\boldsymbol{y})$ where $f : \mathcal{X} \times \mathcal{Y} \to \mathbb{R}$, and $h : \mathcal{X} \times \mathcal{Y} \to \mathbb{R}^d$. Let $\mathcal{Y}^*(\boldsymbol{x}) = \arg\max_{\boldsymbol{y} \in \mathcal{Y}: h(\boldsymbol{x},\boldsymbol{y}) \geq 0} f(\boldsymbol{x},\boldsymbol{y})$. Suppose that 1. $f(\boldsymbol{x},\boldsymbol{y}), h_1(\boldsymbol{x},\boldsymbol{y}), \ldots, h_d(\boldsymbol{x},\boldsymbol{y})$ are continuous in $(\boldsymbol{x},\boldsymbol{y})$ and $f^*$ convex in $\boldsymbol{x}$; 2. $\nabla_{\boldsymbol{x}} f, \nabla_{\boldsymbol{x}} h_1, \ldots, \nabla_{\boldsymbol{x}} h_d$ are continuous in $(\boldsymbol{x},\boldsymbol{y})$; 3. $\mathcal{Y}$ is non-empty and compact, and 4. (Slater's condition) $\forall \boldsymbol{x} \in \mathcal{X}, \exists \widehat{\boldsymbol{y}} \in \mathcal{Y}$ s.t. $g_k(\boldsymbol{x}, \widehat{\boldsymbol{y}}) > 0$, for all $k = 1, \ldots, d$. Then, $f^*$ is subdifferentiable and at any point $\widehat{\boldsymbol{x}} \in \mathcal{X}$, $\partial_{\boldsymbol{x}} f^*(\widehat{\boldsymbol{x}}) =$*

$$\mathrm{conv}\left(\bigcup_{\boldsymbol{y}^*(\widehat{\boldsymbol{x}}) \in \mathcal{Y}^*(\widehat{\boldsymbol{x}})} \bigcup_{\lambda_k^*(\widehat{\boldsymbol{x}}, \boldsymbol{y}^*(\widehat{\boldsymbol{x}})) \in \Lambda^*(\widehat{\boldsymbol{x}}, \boldsymbol{y}^*(\widehat{\boldsymbol{x}}))} \left\{ \nabla_{\boldsymbol{x}} f(\widehat{\boldsymbol{x}}, \boldsymbol{y}^*(\widehat{\boldsymbol{x}})) + \sum_{k=1}^d \lambda_k^*(\widehat{\boldsymbol{x}}, \boldsymbol{y}^*(\widehat{\boldsymbol{x}})) \nabla_{\boldsymbol{x}} g_k(\widehat{\boldsymbol{x}}, \boldsymbol{y}^*(\widehat{\boldsymbol{x}})) \right\} \right),$$
(37)

*where conv is the convex hull operator and $\boldsymbol{\lambda}^*(\widehat{\boldsymbol{x}}, \boldsymbol{y}^*(\widehat{\boldsymbol{x}})) = (\lambda_1^*(\widehat{\boldsymbol{x}}, \boldsymbol{y}^*(\widehat{\boldsymbol{x}})), \ldots, \lambda_d^*(\widehat{\boldsymbol{x}}, \boldsymbol{y}^*(\widehat{\boldsymbol{x}})))^T \in \Lambda^*(\widehat{\boldsymbol{x}}, \boldsymbol{y}^*(\widehat{\boldsymbol{x}}))$ are the optimal KKT multipliers associated with $\boldsymbol{y}^*(\widehat{\boldsymbol{x}}) \in \mathcal{Y}^*(\widehat{\boldsymbol{x}})$.*

**Theorem C.2** (Subdifferential Benveniste-Scheinkman Theorem). *Consider the Bellman equation associated with a recursive stochastic optimization problem where $r : \mathcal{S} \times \mathcal{X} \times \mathcal{Y} \to \mathbb{R}$, with state space $\mathcal{S}$ and parameter set $\mathcal{X}$, and $\gamma \in (0, 1)$:*

$$v(\boldsymbol{s}, \boldsymbol{x}) = \max_{\boldsymbol{y} \in \mathcal{Y}: g(\boldsymbol{s},\boldsymbol{x},\boldsymbol{y}) \geq 0} \left\{ r(\boldsymbol{s}, \boldsymbol{x}, \boldsymbol{y}) + \gamma \mathop{\mathbb{E}}_{\boldsymbol{S}' \sim p(\cdot|\boldsymbol{s},\boldsymbol{x},\boldsymbol{y})} [v(\boldsymbol{S}', \boldsymbol{x})] \right\}$$
(38)

*Suppose that Assumption 1.1 holds, and that 1. for all $\boldsymbol{s} \in \mathcal{S}, \boldsymbol{y} \in \mathcal{Y}$, $r(\boldsymbol{s}, \boldsymbol{x}, \boldsymbol{y})$, $g_1(\boldsymbol{s}, \boldsymbol{x}, \boldsymbol{y}), \ldots, g_d(\boldsymbol{s}, \boldsymbol{x}, \boldsymbol{y})$ are concave in $\boldsymbol{x}$, 2. $\nabla_{\boldsymbol{x}} r(\boldsymbol{s}, \boldsymbol{x}, \boldsymbol{y}), \nabla_{\boldsymbol{x}} g_1(\boldsymbol{s}, \boldsymbol{x}, \boldsymbol{y}), \ldots, \nabla_{\boldsymbol{x}} g_d(\boldsymbol{s}, \boldsymbol{x}, \boldsymbol{y}), \nabla_{\boldsymbol{x}} p(\boldsymbol{s}' \mid \boldsymbol{s}, \boldsymbol{x}, \boldsymbol{y})$ are continuous in $(\boldsymbol{s}, \boldsymbol{s}', \boldsymbol{x}, \boldsymbol{y})$, 3. $v(\boldsymbol{s}, \boldsymbol{x})$ is convex in $\boldsymbol{x}$ 4. Slater's condition holds for the optimization problem, i.e., $\forall \boldsymbol{x} \in \mathcal{X}, \boldsymbol{s} \in \mathcal{S}, \exists \widehat{\boldsymbol{y}} \in \mathcal{Y}$ s.t. $g_k(\boldsymbol{s}, \boldsymbol{x}, \widehat{\boldsymbol{y}}) > 0$, for all $k = 1, \ldots, d$.*
*Let $\mathcal{Y}^*(\boldsymbol{s}, \boldsymbol{x}) = \max_{\boldsymbol{y} \in \mathcal{Y}: g(\boldsymbol{s},\boldsymbol{x},\boldsymbol{y}) \geq 0} \left\{ r(\boldsymbol{s}, \boldsymbol{x}, \boldsymbol{y}) + \gamma \mathbb{E}_{\boldsymbol{s}' \sim p(\cdot|\boldsymbol{s},\boldsymbol{x},\boldsymbol{y})}[v(\boldsymbol{s}', \boldsymbol{x})] \right\}$, then $v$ is subdifferentiable and $\partial_{\boldsymbol{x}} v(\boldsymbol{s}, \hat{\boldsymbol{x}}) =$*

$$\mathrm{conv}\left(\bigcup_{\boldsymbol{y}^*(\boldsymbol{s},\widehat{\boldsymbol{x}}) \in \mathcal{Y}^*(\boldsymbol{s},\widehat{\boldsymbol{x}})} \bigcup_{\lambda_k^*(\boldsymbol{s},\widehat{\boldsymbol{x}},\boldsymbol{y}^*(\boldsymbol{s},\widehat{\boldsymbol{x}})) \in \Lambda^*(\boldsymbol{s},\widehat{\boldsymbol{x}},\boldsymbol{y}^*(\widehat{\boldsymbol{x}}))} \left\{ \nabla_{\boldsymbol{x}} r(\boldsymbol{s}, \widehat{\boldsymbol{x}}, \boldsymbol{y}^*(\boldsymbol{s}, \widehat{\boldsymbol{x}})) + \gamma \nabla_{\boldsymbol{x}} \mathop{\mathbb{E}}_{\boldsymbol{S}' \sim p(\cdot|\boldsymbol{s},\widehat{\boldsymbol{x}},\boldsymbol{y}^*(\boldsymbol{s},\widehat{\boldsymbol{x}}))} [v(\boldsymbol{S}', \hat{\boldsymbol{x}})] \right.\right.$$
$$\left.\left. + \sum_{k=1}^d \lambda_k^*(\boldsymbol{s}, \widehat{\boldsymbol{x}}, \boldsymbol{y}^*(\boldsymbol{s}, \widehat{\boldsymbol{x}})) \nabla_{\boldsymbol{x}} g_k(\boldsymbol{s}, \widehat{\boldsymbol{x}}, \boldsymbol{y}^*(\boldsymbol{s}, \widehat{\boldsymbol{x}})) \right\} \right).$$
(39)

*Suppose additionally, that for all $\boldsymbol{s}, \boldsymbol{s}' \in \mathcal{S}, \boldsymbol{x} \in \mathcal{X}, \boldsymbol{y}^*(\boldsymbol{s}, \boldsymbol{x}) \in \mathcal{Y}^*(\boldsymbol{s}, \boldsymbol{x}) \nabla_{\boldsymbol{x}} p(\boldsymbol{s}' \mid \boldsymbol{s}, \boldsymbol{x}, \boldsymbol{y}^*(\boldsymbol{s}, \boldsymbol{x})) > 0$, then $\partial_{\boldsymbol{x}} v(\boldsymbol{s}, \hat{\boldsymbol{x}}) =$*

$$\mathrm{conv}\left(\bigcup_{\boldsymbol{y}^*(\boldsymbol{s},\widehat{\boldsymbol{x}}) \in \mathcal{Y}^*(\boldsymbol{s},\widehat{\boldsymbol{x}})} \bigcup_{\lambda_k^*(\boldsymbol{s},\widehat{\boldsymbol{x}},\boldsymbol{y}^*(\boldsymbol{s},\widehat{\boldsymbol{x}})) \in \Lambda^*(\boldsymbol{s},\widehat{\boldsymbol{x}},\boldsymbol{y}^*(\widehat{\boldsymbol{x}}))} \left\{ \nabla_{\boldsymbol{x}} r(\boldsymbol{s}, \widehat{\boldsymbol{x}}, \boldsymbol{y}^*(\boldsymbol{s}, \widehat{\boldsymbol{x}})) + \gamma \mathop{\mathbb{E}}_{\boldsymbol{S}' \sim p(\cdot|\boldsymbol{s},\widehat{\boldsymbol{x}},\boldsymbol{y}^*(\boldsymbol{s},\widehat{\boldsymbol{x}}))} [\nabla_{\boldsymbol{x}} v(\boldsymbol{S}', \hat{\boldsymbol{x}})] \right.\right.$$
$$\left.\left. + \gamma \mathop{\mathbb{E}}_{\boldsymbol{S}' \sim p(\cdot|\boldsymbol{s},\widehat{\boldsymbol{x}},\boldsymbol{y}^*(\boldsymbol{s},\widehat{\boldsymbol{x}}))} [v(\boldsymbol{S}', \hat{\boldsymbol{x}}) \nabla_{\boldsymbol{x}} \log(p(\boldsymbol{S}' \mid \boldsymbol{s}, \hat{\boldsymbol{x}}, \boldsymbol{y}^*(\boldsymbol{s}, \widehat{\boldsymbol{x}})))] + \sum_{k=1}^d \lambda_k^*(\boldsymbol{s}, \widehat{\boldsymbol{x}}, \boldsymbol{y}^*(\boldsymbol{s}, \widehat{\boldsymbol{x}})) \nabla_{\boldsymbol{x}} g_k(\boldsymbol{s}, \widehat{\boldsymbol{x}}, \boldsymbol{y}^*(\boldsymbol{s}, \widehat{\boldsymbol{x}})) \right\} \right).$$
(40)

*where conv is the convex hull operator and $\boldsymbol{\lambda}^*(\boldsymbol{s}, \widehat{\boldsymbol{x}}, \boldsymbol{y}^*(\boldsymbol{s}, , \widehat{\boldsymbol{x}})) = (\lambda_1^*(\boldsymbol{s}, \widehat{\boldsymbol{x}}, \boldsymbol{y}^*(\boldsymbol{s}, \widehat{\boldsymbol{x}})), \ldots, \lambda_d^*(\boldsymbol{s}, \widehat{\boldsymbol{x}}, \boldsymbol{y}^*(\boldsymbol{s}, \widehat{\boldsymbol{x}})))^T \in \Lambda^*(\boldsymbol{s}, \widehat{\boldsymbol{x}}, \boldsymbol{y}^*(\boldsymbol{s}, \widehat{\boldsymbol{x}}))$ are the optimal KKT multipliers associated with $\boldsymbol{y}^*(\boldsymbol{s}, \widehat{\boldsymbol{x}}) \in \mathcal{Y}^*(\boldsymbol{s}, \widehat{\boldsymbol{x}})$.*

*Proof of Theorem C.2.* From Theorem C.1, we obtain the first part of the theorem:

$$\partial_{\boldsymbol{x}} v(\boldsymbol{s}, \hat{\boldsymbol{x}}) \tag{41}$$

$$= \text{conv} \left( \bigcup_{\boldsymbol{y}^*(\boldsymbol{s},\hat{\boldsymbol{x}}) \in \mathcal{Y}^*(\boldsymbol{s},\hat{\boldsymbol{x}})} \bigcup_{\lambda_k^*(\boldsymbol{s},\hat{\boldsymbol{x}},\boldsymbol{y}^*(\boldsymbol{s},\hat{\boldsymbol{x}})) \in \Lambda^*(\boldsymbol{s},\hat{\boldsymbol{x}},\boldsymbol{y}^*(\boldsymbol{s},\hat{\boldsymbol{x}}))} \left\{ \nabla_{\boldsymbol{x}} r\left(\boldsymbol{s}, \hat{\boldsymbol{x}}, \boldsymbol{y}^*(\boldsymbol{s}, \hat{\boldsymbol{x}})\right) + \gamma \nabla_{\boldsymbol{x}} \underset{\boldsymbol{s}' \sim p(\cdot|\boldsymbol{s},\boldsymbol{x},\boldsymbol{y}^*(\boldsymbol{s},\hat{\boldsymbol{x}}))}{\mathbb{E}} \left[ v(\boldsymbol{S}', \boldsymbol{x}) \right] \right. \right.$$

$$\left. \left. + \sum_{k=1}^{d} \lambda_k^*(\boldsymbol{s}, \hat{\boldsymbol{x}}, \boldsymbol{y}^*(\boldsymbol{s}, \hat{\boldsymbol{x}})) \nabla_{\boldsymbol{x}} g_k\left(\boldsymbol{s}, \hat{\boldsymbol{x}}, \boldsymbol{y}^*(\boldsymbol{s}, \hat{\boldsymbol{x}})\right) \right\} \right) . \tag{42}$$

By the Leibniz integral rule [67], the gradient of the expectation can instead be expressed as an expectation of the gradient under continuity of the function whose expectation is taken, in this case $v$. In particular, if for all $\boldsymbol{s}, \boldsymbol{s}' \in \mathcal{S}, \boldsymbol{x} \in \mathcal{X}, \boldsymbol{y}^*(\boldsymbol{s}, \boldsymbol{x}) \in \mathcal{Y}^*(\boldsymbol{s}, \boldsymbol{x}) \nabla_{\boldsymbol{x}} p(\boldsymbol{s}' \mid \boldsymbol{s}, \boldsymbol{x}, \boldsymbol{y}^*(\boldsymbol{s}, \boldsymbol{x})) > 0$ we have:

$$\nabla_{\boldsymbol{x}} \underset{\boldsymbol{S}' \sim p(\cdot|\boldsymbol{s},\boldsymbol{x},\boldsymbol{y})}{\mathbb{E}} \left[ v(\boldsymbol{S}', \boldsymbol{x}) \right] \tag{43}$$

$$= \nabla_{\boldsymbol{x}} \int_{\boldsymbol{z} \in \mathcal{S}} p(\boldsymbol{z} \mid \boldsymbol{s}, \boldsymbol{x}, \boldsymbol{y}) v(\boldsymbol{z}, \boldsymbol{x}) d\boldsymbol{z} \tag{44}$$

$$= \int_{\boldsymbol{z} \in \mathcal{S}} \nabla_{\boldsymbol{x}} [p(\boldsymbol{z} \mid \boldsymbol{s}, \boldsymbol{x}, \boldsymbol{y}) v(\boldsymbol{z}, \boldsymbol{x})] d\boldsymbol{z} \qquad \text{(Leibniz Integral Rule)}$$
$$\tag{45}$$

$$= \int_{\boldsymbol{z} \in \mathcal{S}} [p(\boldsymbol{z} \mid \boldsymbol{s}, \boldsymbol{x}, \boldsymbol{y}) \nabla_{\boldsymbol{x}} v(\boldsymbol{z}, \boldsymbol{x}) + v(\boldsymbol{z}, \boldsymbol{x}) \nabla_{\boldsymbol{x}} p(\boldsymbol{z} \mid \boldsymbol{s}, \boldsymbol{x}, \boldsymbol{y})] d\boldsymbol{z} \qquad \text{(Product Rule)}$$
$$\tag{46}$$

$$= \int_{\boldsymbol{z} \in \mathcal{S}} \left[ p(\boldsymbol{z} \mid \boldsymbol{s}, \boldsymbol{x}, \boldsymbol{y}) \nabla_{\boldsymbol{x}} v(\boldsymbol{z}, \boldsymbol{x}) + v(\boldsymbol{z}, \boldsymbol{x}) p(\boldsymbol{z} \mid \boldsymbol{s}, \boldsymbol{x}, \boldsymbol{y}) \frac{\nabla_{\boldsymbol{x}} p(\boldsymbol{z} \mid \boldsymbol{s}, \boldsymbol{x}, \boldsymbol{y})}{p(\boldsymbol{z} \mid \boldsymbol{s}, \boldsymbol{x}, \boldsymbol{y})} \right] d\boldsymbol{z} \qquad (p(\boldsymbol{z} \mid \boldsymbol{s}, \boldsymbol{x}, \boldsymbol{y}) > 0)$$
$$\tag{47}$$

$$= \int_{\boldsymbol{z} \in \mathcal{S}} [p(\boldsymbol{z} \mid \boldsymbol{s}, \boldsymbol{x}, \boldsymbol{y}) \nabla_{\boldsymbol{x}} v(\boldsymbol{z}, \boldsymbol{x}) + v(\boldsymbol{z}, \boldsymbol{x}) p(\boldsymbol{z} \mid \boldsymbol{s}, \boldsymbol{x}, \boldsymbol{y}) \nabla_{\boldsymbol{x}} \log p(\boldsymbol{z} \mid \boldsymbol{s}, \boldsymbol{x}, \boldsymbol{y})] d\boldsymbol{z} \tag{48}$$

$$= \int_{\boldsymbol{z} \in \mathcal{S}} [p(\boldsymbol{z} \mid \boldsymbol{s}, \boldsymbol{x}, \boldsymbol{y}) \nabla_{\boldsymbol{x}} v(\boldsymbol{z}, \boldsymbol{x})] d\boldsymbol{z} + \int_{\boldsymbol{z} \in \mathcal{S}} [v(\boldsymbol{z}, \boldsymbol{x}) p(\boldsymbol{z} \mid \boldsymbol{s}, \boldsymbol{x}, \boldsymbol{y}) \nabla_{\boldsymbol{x}} \log p(\boldsymbol{z} \mid \boldsymbol{s}, \boldsymbol{x}, \boldsymbol{y})] d\boldsymbol{z}$$
$$\tag{49}$$

$$= \underset{\boldsymbol{S}' \sim p(\cdot|\boldsymbol{s},\boldsymbol{x},\boldsymbol{y})}{\mathbb{E}} \left[ \nabla_{\boldsymbol{x}} v(\boldsymbol{S}', \boldsymbol{x}) \right] + \underset{\boldsymbol{S}' \sim p(\cdot|\boldsymbol{s},\boldsymbol{x},\boldsymbol{y})}{\mathbb{E}} \left[ v(\boldsymbol{S}', \boldsymbol{x}) \nabla_{\boldsymbol{x}} \log p(\boldsymbol{S}' \mid \boldsymbol{s}, \boldsymbol{x}, \boldsymbol{y}) \right] \tag{50}$$

This gives us $\partial_{\boldsymbol{x}} v(\boldsymbol{s}, \hat{\boldsymbol{x}}) =$

$$\text{conv} \left( \bigcup_{\boldsymbol{y}^*(\boldsymbol{s},\hat{\boldsymbol{x}}) \in \mathcal{Y}^*(\boldsymbol{s},\hat{\boldsymbol{x}})} \bigcup_{\lambda_k^*(\boldsymbol{s},\hat{\boldsymbol{x}},\boldsymbol{y}^*(\boldsymbol{s},\hat{\boldsymbol{x}})) \in \Lambda^*(\boldsymbol{s},\hat{\boldsymbol{x}},\boldsymbol{y}^*(\hat{\boldsymbol{x}}))} \left\{ \nabla_{\boldsymbol{x}} r\left(\boldsymbol{s}, \hat{\boldsymbol{x}}, \boldsymbol{y}^*(\boldsymbol{s}, \hat{\boldsymbol{x}})\right) + \gamma \underset{\boldsymbol{S}' \sim p(\cdot|\boldsymbol{s},\hat{\boldsymbol{x}},\boldsymbol{y}^*(\boldsymbol{s},\hat{\boldsymbol{x}}))}{\mathbb{E}} \left[ \nabla_{\boldsymbol{x}} v(\boldsymbol{S}', \hat{\boldsymbol{x}}) \right] \right. \right.$$

$$\left. \left. + \gamma \underset{\boldsymbol{S}' \sim p(\cdot|\boldsymbol{s},\hat{\boldsymbol{x}},\boldsymbol{y}^*(\boldsymbol{s},\hat{\boldsymbol{x}}))}{\mathbb{E}} \left[ v(\boldsymbol{S}', \hat{\boldsymbol{x}}) \nabla_{\boldsymbol{x}} \log \left( p(\boldsymbol{S}' \mid \boldsymbol{s}, \hat{\boldsymbol{x}}, \boldsymbol{y}^*(\boldsymbol{s}, \hat{\boldsymbol{x}})) \right) \right] + \sum_{k=1}^{d} \lambda_k^*(\boldsymbol{s}, \hat{\boldsymbol{x}}, \boldsymbol{y}^*(\boldsymbol{s}, \hat{\boldsymbol{x}})) \nabla_{\boldsymbol{x}} g_k\left(\boldsymbol{s}, \hat{\boldsymbol{x}}, \boldsymbol{y}^*(\boldsymbol{s}, \hat{\boldsymbol{x}})\right) \right\} \right) .$$
$$\tag{51}$$

$\square$

Note that in the special case that the probability transition function is representing a deterministic recursive parametrized optimization problem, $v(\boldsymbol{s}, \boldsymbol{x}) = \max_{\boldsymbol{y} \in \mathcal{Y}: \boldsymbol{g}(\boldsymbol{s},\boldsymbol{x},\boldsymbol{y}) \geq 0} \{ r(\boldsymbol{s}, \boldsymbol{x}, \boldsymbol{y}) + \gamma [v(\tau(\boldsymbol{s}, \boldsymbol{x}, \boldsymbol{y}), \boldsymbol{x})] \}$ i.e., $p(\boldsymbol{s}' \mid \boldsymbol{s}, \boldsymbol{x}, \boldsymbol{y}) \in \{0, 1\}$ for all $\boldsymbol{s}, \boldsymbol{s}' \in \mathcal{S}, \boldsymbol{x} \in \mathcal{X}, \boldsymbol{y} \in \mathcal{Y}$, and $\tau : \mathcal{S} \times \mathcal{X} \times \mathcal{Y} \to \mathcal{S}$ is such that $\tau(\boldsymbol{s}, \boldsymbol{x}, \boldsymbol{y}) = \boldsymbol{s}'$ iff $p(\boldsymbol{s}' \mid \boldsymbol{s}, \boldsymbol{x}, \boldsymbol{y}) = 1$, the CSD convexity assumption reduces to the linearity of the deterministic state

transition function $\tau$ (Proposition 1 of [68]). In this case, the subdifferential of the Bellman equation reduces to

$$\partial_{\boldsymbol{x}} v(\boldsymbol{s}, \hat{\boldsymbol{x}}) \tag{52}$$

$$= \text{conv}\left(\bigcup_{\boldsymbol{y}^*(\boldsymbol{s}, \hat{\boldsymbol{x}}) \in \mathcal{Y}^*(\boldsymbol{s}, \hat{\boldsymbol{x}})} \bigcup_{\lambda_k^*(\boldsymbol{s}, \hat{\boldsymbol{x}}, \boldsymbol{y}^*(\boldsymbol{s}, \hat{\boldsymbol{x}})) \in \Lambda^*(\boldsymbol{s}, \hat{\boldsymbol{x}}, \boldsymbol{y}^*(\boldsymbol{s}, \hat{\boldsymbol{x}}))} \left\{ \nabla_{\boldsymbol{x}} r\left(\boldsymbol{s}, \hat{\boldsymbol{x}}, \boldsymbol{y}^*(\boldsymbol{s}, \hat{\boldsymbol{x}})\right) + \gamma \nabla_{\boldsymbol{x}} \tau(\boldsymbol{s}, \hat{\boldsymbol{x}}, \boldsymbol{y}) \nabla_{\boldsymbol{s}} v(\tau(\boldsymbol{s}, \hat{\boldsymbol{x}}, \boldsymbol{y}), \hat{\boldsymbol{x}}) \right.$$

$$\left. \left. + \nabla_{\boldsymbol{x}} v(\tau(\boldsymbol{s}, \hat{\boldsymbol{x}}), \hat{\boldsymbol{x}}) + \sum_{k=1}^{d} \lambda_k^*(\boldsymbol{s}, \hat{\boldsymbol{x}}, \boldsymbol{y}^*(\boldsymbol{s}, \hat{\boldsymbol{x}})) \nabla_{\boldsymbol{x}} g_k\left(\boldsymbol{s}, \hat{\boldsymbol{x}}, \boldsymbol{y}^*(\boldsymbol{s}, \hat{\boldsymbol{x}})\right) \right\} \right) \tag{53}$$

$$\tag{54}$$

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

*Proof of Theorem 3.1.* By Theorem 2.1 and Theorem 2.3 we know that $(\boldsymbol{\pi}_{\boldsymbol{x}}^*, \boldsymbol{\pi}_{\boldsymbol{y}}^*)$ is a recursive Stackelberg equilibrium iff

$$v^{\boldsymbol{\pi}_{\boldsymbol{y}} \boldsymbol{\pi}_{\boldsymbol{y}}^*}(\boldsymbol{s}) = \left(Cv^{\boldsymbol{\pi}_{\boldsymbol{y}} \boldsymbol{\pi}_{\boldsymbol{y}}^*}\right)(\boldsymbol{s}) \ . \tag{55}$$

Note that for any policy profile $(\boldsymbol{\pi}_{\boldsymbol{x}}^*, \boldsymbol{\pi}_{\boldsymbol{y}}^*)$ that satisfies $v^{\boldsymbol{\pi}_{\boldsymbol{y}} \boldsymbol{\pi}_{\boldsymbol{y}}^*}(\boldsymbol{s}) = \left(Cv^{\boldsymbol{\pi}_{\boldsymbol{y}} \boldsymbol{\pi}_{\boldsymbol{y}}^*}\right)(\boldsymbol{s})$ by Definition 1.2 we have that $(\boldsymbol{\pi}_{\boldsymbol{x}}^*(\boldsymbol{s}), \boldsymbol{\pi}_{\boldsymbol{y}}^*(\boldsymbol{s}))$ is a Stackelberg equilibrium of

$$\min_{\boldsymbol{x} \in \mathcal{X}} \max_{\boldsymbol{y} \in \mathcal{Y} : g(\boldsymbol{s}, \boldsymbol{x}, \boldsymbol{y}) \ge \boldsymbol{0}} \left\{ r(\boldsymbol{s}, \boldsymbol{x}, \boldsymbol{y}) + \gamma \mathbb{E}_{\boldsymbol{S}' \sim p(\cdot \mid \boldsymbol{s}, \boldsymbol{x}, \boldsymbol{y})} [v(\boldsymbol{S}')] \right\}$$

for all $\boldsymbol{s} \in \mathcal{S}$.

Fix a state $\boldsymbol{s} \in \mathcal{S}$, under the assumptions of the theorem, the conditions of Theorem C.2 are satisfied and $u^*(\boldsymbol{s}, \boldsymbol{x}) = \max_{\boldsymbol{y} \in \mathcal{Y} : g(\boldsymbol{s}, \boldsymbol{x}, \boldsymbol{y}) \ge \boldsymbol{0}} \{r(\boldsymbol{s}, \boldsymbol{x}, \boldsymbol{y}) + \gamma \mathbb{E}_{\boldsymbol{S}' \sim p(\cdot \mid \boldsymbol{s}, \boldsymbol{x}, \boldsymbol{y})} [v(\boldsymbol{S}')]\}$ is subdifferentiable in $\boldsymbol{x}$. Since $u^*(\boldsymbol{s}, \boldsymbol{x})$ is convex in $\boldsymbol{x}$, and Slater's condition is satisfied by the assumptions of the theorem, the necessary *and sufficient* conditions for $\boldsymbol{\pi}_{\boldsymbol{x}}^*(\boldsymbol{s})$ to be an optimal solution to $\min_{\boldsymbol{x} \in \mathcal{X}} u^*(\boldsymbol{s}, \boldsymbol{x})$ are given by the KKT conditions [69] for $\min_{\boldsymbol{x} \in \mathcal{X}} u^*(\boldsymbol{s}, \boldsymbol{x})$. Note that we can state the first order KKT conditions explicitly thanks to the subdifferential Benveniste-Scheinkman theorem (Theorem C.2). That is, $\boldsymbol{\pi}_{\boldsymbol{x}}^*(\boldsymbol{s})$ is an optimal solution to $\min_{\boldsymbol{x} \in \mathcal{X}} u^*(\boldsymbol{s}, \boldsymbol{x})$ if there exists $\boldsymbol{\mu}^*(\boldsymbol{s}) \in \mathbb{R}_+^p$ such that:

$$\nabla_{\boldsymbol{x}}\mathcal{L}_{\boldsymbol{s},\boldsymbol{\pi}_{\boldsymbol{x}}^*(\boldsymbol{s})}(\boldsymbol{y}^*(\boldsymbol{s},\boldsymbol{\pi}_{\boldsymbol{x}}^*(\boldsymbol{s})),\boldsymbol{\lambda}^*(\boldsymbol{s},\boldsymbol{\pi}_{\boldsymbol{x}}^*(\boldsymbol{s}),\boldsymbol{y}^*(\boldsymbol{s},\boldsymbol{\pi}_{\boldsymbol{x}}^*(\boldsymbol{s})))+\sum_{k=1}^{p}\mu_k^*(\boldsymbol{s})\nabla_{\boldsymbol{x}}q_k(\boldsymbol{\pi}_{\boldsymbol{x}}^*(\boldsymbol{s}))=0 \tag{56}$$

$$\mu_k^*(\boldsymbol{s})q_k(\boldsymbol{\pi}_{\boldsymbol{x}}^*(\boldsymbol{s}))=0 \qquad\qquad \forall k\in[p] \tag{57}$$

$$q_k(\boldsymbol{\pi}_{\boldsymbol{x}}^*(\boldsymbol{s}))\le 0 \qquad\qquad \forall k\in[p] \tag{58}$$

where $\boldsymbol{y}^*(\boldsymbol{s},\boldsymbol{\pi}_{\boldsymbol{x}}^*(\boldsymbol{s}))\in\arg\max_{\boldsymbol{y}\in\mathcal{Y}:g(\boldsymbol{s},\boldsymbol{\pi}_{\boldsymbol{x}}^*(\boldsymbol{s}),\boldsymbol{y})\ge\mathbf{0}}\left\{r(\boldsymbol{s},\boldsymbol{\pi}_{\boldsymbol{x}}^*(\boldsymbol{s}),\boldsymbol{y})+\gamma\,\mathbb{E}_{\boldsymbol{S}'\sim p(\cdot|\boldsymbol{s},\boldsymbol{\pi}_{\boldsymbol{x}}^*(\boldsymbol{s}),\boldsymbol{y})}[v(\boldsymbol{S}')]\right\}$
and
$\boldsymbol{\lambda}^*(\boldsymbol{s},\boldsymbol{\pi}_{\boldsymbol{x}}^*(\boldsymbol{s}),\boldsymbol{y}^*(\boldsymbol{s},\boldsymbol{\pi}_{\boldsymbol{x}}^*(\boldsymbol{s})))=(\lambda_1^*(\boldsymbol{s},\boldsymbol{\pi}_{\boldsymbol{x}}^*(\boldsymbol{s}),\boldsymbol{y}^*(\boldsymbol{s},\boldsymbol{\pi}_{\boldsymbol{x}}^*(\boldsymbol{s}))),\ldots,\lambda_d^*(\boldsymbol{s},\boldsymbol{\pi}_{\boldsymbol{x}}^*(\boldsymbol{s}),\boldsymbol{y}^*(\boldsymbol{s},\boldsymbol{\pi}_{\boldsymbol{x}}^*(\boldsymbol{s}))))^T\in$
$\Lambda^*(\boldsymbol{s},\boldsymbol{\pi}_{\boldsymbol{x}}^*(\boldsymbol{s}),\boldsymbol{y}^*(\boldsymbol{s},\boldsymbol{\pi}_{\boldsymbol{x}}^*(\boldsymbol{s})))$ are the optimal KKT multipliers associated with $\boldsymbol{y}^*(\boldsymbol{s},\boldsymbol{\pi}_{\boldsymbol{x}}^*(\boldsymbol{s}))\in$
$\mathcal{Y}^*(\boldsymbol{s},\boldsymbol{\pi}_{\boldsymbol{x}}^*(\boldsymbol{s}))$ which are guaranteed to exist since Slater's condition is satisfied for
$\max_{\boldsymbol{y}\in\mathcal{Y}:g(\boldsymbol{s},\boldsymbol{x},\boldsymbol{y})\ge\mathbf{0}}\left\{r(\boldsymbol{s},\boldsymbol{x},\boldsymbol{y})+\gamma\,\mathbb{E}_{\boldsymbol{S}'\sim p(\cdot|\boldsymbol{s},\boldsymbol{x},\boldsymbol{y})}[v(\boldsymbol{S}')]\right\}$.

Similarly, fix a state $\boldsymbol{s}\in\mathcal{S}$ and an action for the outer player $\boldsymbol{x}\in\mathcal{X}$, since Slater's condition is satisfied for $\max_{\boldsymbol{y}\in\mathcal{Y}:g(\boldsymbol{s},\boldsymbol{x},\boldsymbol{y})\ge\mathbf{0}}\left\{r(\boldsymbol{s},\boldsymbol{x},\boldsymbol{y})+\gamma\,\mathbb{E}_{\boldsymbol{S}'\sim p(\cdot|\boldsymbol{s},\boldsymbol{x},\boldsymbol{y})}[v(\boldsymbol{S}')]\right\}$, the necessary conditions for $\boldsymbol{\pi}_{\boldsymbol{y}}^*(\boldsymbol{s})$ to be a Stackelberg equilibrium strategy for the inner player at state $\boldsymbol{s}$ are given by the KKT conditions for $\max_{\boldsymbol{y}\in\mathcal{Y}:g(\boldsymbol{s},\boldsymbol{x},\boldsymbol{y})\ge\mathbf{0}}\left\{r(\boldsymbol{s},\boldsymbol{x},\boldsymbol{y})+\gamma\,\mathbb{E}_{\boldsymbol{S}'\sim p(\cdot|\boldsymbol{s},\boldsymbol{x},\boldsymbol{y})}[v(\boldsymbol{S}')]\right\}$. That is, there exists $\boldsymbol{\lambda}^*(\boldsymbol{s},\boldsymbol{x})\in\mathbb{R}_+^d$ and $\boldsymbol{\nu}^*(\boldsymbol{s},\boldsymbol{x})\in\mathbb{R}_+^l$ such that:

$$\nabla_{\boldsymbol{y}}\mathcal{L}_{\boldsymbol{s},\boldsymbol{x}}(\boldsymbol{\pi}_{\boldsymbol{y}}^*(\boldsymbol{s}),\boldsymbol{\lambda}^*(\boldsymbol{s},\boldsymbol{x}))+\sum_{k=1}^{l}\nu_k^*(\boldsymbol{s})\nabla_{\boldsymbol{x}}r_k(\boldsymbol{\pi}_{\boldsymbol{y}}^*(\boldsymbol{s}))=0 \tag{59}$$

$$g_k(\boldsymbol{s},\boldsymbol{x},\boldsymbol{\pi}_{\boldsymbol{y}}^*(\boldsymbol{s}))\ge 0 \qquad\qquad \forall k\in[d] \tag{60}$$

$$\lambda_k^*(\boldsymbol{s},\boldsymbol{x})g_k(\boldsymbol{s},\boldsymbol{x},\boldsymbol{\pi}_{\boldsymbol{y}}^*(\boldsymbol{s}))=0 \qquad\qquad \forall k\in[d] \tag{61}$$

$$\nu_k^*(\boldsymbol{s},\boldsymbol{x})\nabla_{\boldsymbol{x}}r_k(\boldsymbol{\pi}_{\boldsymbol{y}}^*(\boldsymbol{s}))=0 \qquad\qquad \forall k\in[l] \tag{62}$$

$$r_k(\boldsymbol{x})\ge 0 \qquad\qquad \forall k\in[l] \tag{63}$$

Combining the necessary and *sufficient* conditions in Equations (56) to (58) with the necessary conditions in Equations (59) to (63), we obtain the necessary conditions for $(\boldsymbol{\pi}_{\boldsymbol{x}}^*,\boldsymbol{\pi}_{\boldsymbol{y}}^*)$ to be a recursive Stackelberg equilibrium.

$\square$

If additionally, the objective of the inner player at each state $\boldsymbol{s}\in\mathcal{S}$, $r(\boldsymbol{s},\boldsymbol{x},\boldsymbol{y})+\gamma\,\mathbb{E}_{\boldsymbol{S}'\sim p(\cdot|\boldsymbol{s},\boldsymbol{x},\boldsymbol{y})}[v(\boldsymbol{S}',\boldsymbol{x})]$ is concave in $\boldsymbol{y}$, then the above conditions become necessary and sufficient. The proof follows exactly the same, albeit the optimality conditions on the inner player's policy become necessary and sufficient.

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

# D  Omitted Results and Proofs Section 4

Before, we introduce the stochastic Stackelberg game whose recursive Stackelberg equilibria correspond to recursive competitive equilibria of an associated stochastic Fisher market, we introduce the following technical lemma, which provides the necessary and sufficient conditions for an allocation and saving system of a buyer to be expected utility maximizing.

**Lemma D.1.** *Consider a stochastic Fisher market* $\mathcal{F}$ *such that the transition probability function* $p$ *is continuous in* $\beta_i$ *and independent of* $\boldsymbol{x}_i$. *For any price system* $\boldsymbol{p} \in \mathbb{R}_+^{\mathcal{S} \times m}$, *a tuple* $(\boldsymbol{x}_i^*, \beta_i^*) \in \mathbb{R}_+^{\mathcal{S} \times m} \times \mathbb{R}_+^m$ *consisting of an allocation system and saving system for a buyer* $i \in [n]$, *given by a continuous, and homogeneous utility function* $u_i : \mathbb{R}_+^m \times \mathcal{T} \to \mathbb{R}$ *representing a locally non-satiated preference, is expected utility maximizing constrained by the saving and spending constrains, i.e.,* $(\boldsymbol{x}_i^*, \beta_i^*)$ *is the optimal policy solving the following recursive Bellman equation* $\nu_i(\boldsymbol{s}) = \max_{(\boldsymbol{x}_i, \beta_i) \in \mathbb{R}_+^{m+1}: \boldsymbol{x}_i \cdot \boldsymbol{p}^*(\boldsymbol{s}) + \beta_i \leq b_i} \left\{ u_i(\boldsymbol{x}_i, \boldsymbol{t}_i) + \gamma \mathbb{E}_{(\boldsymbol{T}', \boldsymbol{b}', \boldsymbol{q}') \sim p(\cdot | \boldsymbol{s}, (\beta_i, \boldsymbol{\beta}_{-i}^*(\boldsymbol{s})))} [\nu_i(\boldsymbol{T}', \boldsymbol{b}' + \beta_i, \boldsymbol{q}')] \right\}$, *only if we have for all states* $\boldsymbol{s} \in \mathcal{S}$, $\boldsymbol{x}_i^*(\boldsymbol{T}, \boldsymbol{b}, \boldsymbol{q}) \cdot \boldsymbol{p}(\boldsymbol{T}, \boldsymbol{b}, \boldsymbol{q}) + \beta_i^*(\boldsymbol{T}, \boldsymbol{b}, \boldsymbol{q}) \leq b_i$, *and,*

$$x_{ij}^*(\boldsymbol{s}) > 0 \implies \frac{\frac{\partial u_i}{\partial x_{ij}}(\boldsymbol{x}_i^*(\boldsymbol{s}); \boldsymbol{t}_i)}{p_j(\boldsymbol{s})} = \frac{u_i(\boldsymbol{x}_i^*(\boldsymbol{s}); \boldsymbol{t}_i)}{b_i - \beta_i^*(\boldsymbol{s})} \qquad \forall j \in [m] \tag{69}$$

$$\beta_i^*(\boldsymbol{s}) > 0 \implies \frac{\partial \nu_i^*}{\partial b_i}(\boldsymbol{s}) = \gamma \frac{\partial}{\partial \beta_i} \mathbb{E}_{(\boldsymbol{T}', \boldsymbol{b}', \boldsymbol{q}')} [\nu_i^*(\boldsymbol{T}', \boldsymbol{b}' + \beta_i^*(\boldsymbol{s}), \boldsymbol{q}')] \tag{70}$$

*If additionally,* $u_i$ *is concave and* $p$ *is CSD concave in* $(\boldsymbol{x}_i, \beta_i)$, *then the above condition becomes also sufficient.*

*Proof of Lemma D.1.* Fix a buyer $i \in [n]$. Throughout we use $\boldsymbol{b} + \beta_i$ as shortcut for $\boldsymbol{b} + (\beta_i, \boldsymbol{0}_{-i})$. Suppose that $\nu_i^*$ solves the following recursive Bellman equation:

$$\nu_i(\boldsymbol{s}) = \max_{(\boldsymbol{x}_i, \beta_i) \in \mathbb{R}_+^{m+1}: \boldsymbol{x}_i \cdot \boldsymbol{p}^*(\boldsymbol{s}) + \beta_i \leq b_i} \left\{ u_i(\boldsymbol{x}_i, \boldsymbol{t}_i) + \gamma \mathbb{E}_{(\boldsymbol{T}', \boldsymbol{b}', \boldsymbol{q}') \sim p(\cdot | \boldsymbol{s}, (\beta_i, \boldsymbol{\beta}_{-i}^*(\boldsymbol{s})))} [\nu_i(\boldsymbol{T}', \boldsymbol{b}' + \beta_i, \boldsymbol{q}')] \right\} \tag{71}$$

Define the Lagrangian associated with the consumption-saving problem:

$$\max_{(\boldsymbol{x}_i, \beta_i) \in \mathbb{R}_+^{m+1}: \boldsymbol{x}_i \cdot \boldsymbol{p}^*(\boldsymbol{s}) + \beta_i \leq b_i} \left\{ u_i(\boldsymbol{x}_i, \boldsymbol{t}_i) + \gamma \mathbb{E}_{(\boldsymbol{T}', \boldsymbol{b}', \boldsymbol{q}') \sim p(\cdot | \boldsymbol{s}, (\beta_i, \boldsymbol{\beta}_{-i}^*(\boldsymbol{s})))} [\nu_i^*(\boldsymbol{T}', \boldsymbol{b}' + \beta_i, \boldsymbol{q}')] \right\} \tag{72}$$

as follows:

$$\mathcal{L}(\boldsymbol{x}_i, \beta_i, \lambda, \boldsymbol{\mu}; \boldsymbol{p}) = u_i(\boldsymbol{x}_i; \boldsymbol{t}_i) + \gamma \mathbb{E}_{(\boldsymbol{T}', \boldsymbol{b}', \boldsymbol{q}') \sim p(\cdot | \boldsymbol{T}, \boldsymbol{b}, \boldsymbol{q}, (\beta_i, \boldsymbol{\beta}_{-i}^*(\boldsymbol{s})))} [\nu_i^*(\boldsymbol{T}', \boldsymbol{b}' + \beta_i, \boldsymbol{q}')]$$

$$+ \lambda(b_i - \boldsymbol{x}_i \boldsymbol{p}) + \sum_{j \in [m]} \mu_j x_{ij} + \mu_{m+1} \beta_i \tag{73}$$

$$\nu_i^*(\boldsymbol{T}, \boldsymbol{b}, \boldsymbol{q}) = \max_{(\boldsymbol{x}_i, \beta_i) \in \mathbb{R}_+^{m+1} : \boldsymbol{x}_i \cdot \boldsymbol{p}^*(\boldsymbol{b}) + \beta_i \leq b_i} u_i(\boldsymbol{x}_i, \boldsymbol{t}_i) + \gamma \underset{(\boldsymbol{T}', \boldsymbol{b}', \boldsymbol{q}')}{\mathbb{E}} [\nu_i^*(\boldsymbol{T}', \boldsymbol{b}' + \beta_i, \boldsymbol{q}')] .$$

Assume that for any state $\boldsymbol{s} \in \mathcal{S}$, we have $b_i > 0$. We can ignore states such that $b_i > 0$ since at those states the buyer cannot be allocated goods, and can also not put aside savings. Then, Slater's condition holds and the necessary first order optimality conditions for an allocation $\boldsymbol{x}_i^*(\boldsymbol{s}) \in \mathbb{R}_+^m$, saving $\beta_i^*(\boldsymbol{s}) \in \mathbb{R}_+$ and associated Lagrangian multipliers $\lambda^*(\boldsymbol{s}) \in \mathbb{R}_+$, $\boldsymbol{\mu}^*(\boldsymbol{s}) \in \mathbb{R}^{m+1}$ to be optimal for any prices $\boldsymbol{p}(\boldsymbol{s}) \in \mathbb{R}_+^m$ and state $\boldsymbol{s} \in \mathcal{S}$ are given by the following pair of KKT conditions [69] for all $j \in [m]$:

$$\frac{\partial u_i}{\partial x_{ij}}(\boldsymbol{x}_i^*(\boldsymbol{s}); \boldsymbol{t}_i) - \lambda^*(\boldsymbol{s}) p_j(\boldsymbol{s}) + \mu_j^*(\boldsymbol{s}) \doteq 0 \tag{74}$$

$$\gamma \frac{\partial}{\partial \beta_i} \underset{(\boldsymbol{T}', \boldsymbol{b}', \boldsymbol{q}')}{\mathbb{E}} [\nu_i^*(\boldsymbol{T}', \boldsymbol{b}' + \beta_i^*, \boldsymbol{q}')] - \lambda_i^*(\boldsymbol{s}) + \mu_{m+1}^*(\boldsymbol{s}) \doteq 0 \tag{75}$$

Additionally, by the KKT complimentarity conditions, we have for all $j \in j$, $\mu_j^* x_{ij}^* = 0$ and $\mu_{m+1}^* \beta_i^* = 0$, which gives us:

$$\frac{\partial u_i}{\partial x_{ij}}(\boldsymbol{x}_i^*(\boldsymbol{s}); \boldsymbol{t}_i) - \lambda^*(\boldsymbol{s}) p_j(\boldsymbol{s}) = 0 \qquad \forall j \in [m] \tag{76}$$

$$\beta_i^*(\boldsymbol{s}) > 0 \implies \gamma \frac{\partial}{\partial \beta_i} \underset{(\boldsymbol{T}', \boldsymbol{b}', \boldsymbol{q}')}{\mathbb{E}} [\nu_i^*(\boldsymbol{T}', \boldsymbol{b}' + \beta_i^*, \boldsymbol{q}')] - \lambda^*(\boldsymbol{s}) = 0 \qquad \forall j \in [m] \tag{77}$$

Re-organizing expressions, yields:

$$x_{ij}^*(\boldsymbol{s}) > 0 \implies \lambda^* = \frac{\frac{\partial u_i}{\partial x_{ij}}(\boldsymbol{x}_i^*(\boldsymbol{s}); \boldsymbol{t}_i)}{p_j(\boldsymbol{s})} \qquad \forall j \in [m] \tag{78}$$

$$\beta_i^*(\boldsymbol{s}) > 0 \implies \lambda^*(\boldsymbol{s}) = \gamma \frac{\partial}{\partial \beta_i} \underset{(\boldsymbol{T}', \boldsymbol{b}', \boldsymbol{q}')}{\mathbb{E}} [\nu_i^*(\boldsymbol{T}', \boldsymbol{b}' + \beta_i^*, \boldsymbol{q}')] \tag{79}$$

Using the envelope theorem [70, 71], we can also compute $\frac{\partial \nu_i^*}{\partial \beta_i}(\boldsymbol{s})$ as follows:

$$\frac{\partial \nu_i^*}{\partial b_i}(\boldsymbol{s}) = \lambda^*(\boldsymbol{s}) \tag{80}$$

We note that for all states $\boldsymbol{s} \in \mathcal{S}$, $\frac{\partial \nu_i^*}{\partial b_i}(\boldsymbol{s})$ is well-defined since $\lambda^*(\boldsymbol{s})$ is uniquely defined for all states by Equation (78). Hence, combining the above with Equation (78) and Equation (79), we get:

$$x_{ij}^*(\boldsymbol{s}) > 0 \implies \lambda^* = \frac{\frac{\partial u_i}{\partial x_{ij}}(\boldsymbol{x}_i^*(\boldsymbol{s}); \boldsymbol{t}_i)}{p_j} \qquad \forall j \in [m] \tag{81}$$

$$\beta_i^*(\boldsymbol{s}) > 0 \implies \frac{\partial \nu_i^*}{\partial b_i}(\boldsymbol{s}) = \gamma \frac{\partial}{\partial \beta_i} \underset{(\boldsymbol{T}', \boldsymbol{b}', \boldsymbol{q}')}{\mathbb{E}} [\nu_i^*(\boldsymbol{T}', \boldsymbol{b}' + \beta_i^*, \boldsymbol{q}')] \tag{82}$$

Finally, going back to Equation (74), multiplying by $x_{ij}^*(\boldsymbol{s})$ and summing up across all $j \in [m]$, we obtain:

$$\sum_{j \in [m]} x_{ij}^*(\boldsymbol{s}) \frac{\partial u_i}{\partial x_{ij}}(\boldsymbol{x}_i^*(\boldsymbol{s}); \boldsymbol{t}_i) - \lambda^*(\boldsymbol{s}) p_j(\boldsymbol{s}) x_{ij}^*(\boldsymbol{s}) + \mu_j^* x_{ij}^*(\boldsymbol{s}) = 0 \tag{83}$$

Using Euler's theorem for homogeneous functions on the partial derivatives of the utility functions, we then have:

$$u_i(\boldsymbol{x}_i^*(\boldsymbol{s}); \boldsymbol{t}_i) - \lambda^*(\boldsymbol{s}) \sum_{j \in [m]} p_j(\boldsymbol{s}) x_{ij}^*(\boldsymbol{s}) + \mu_j^* x_{ij}^*(\boldsymbol{s}) = 0 \tag{84}$$

Additionally, the KKT Slack complementarity conditions, we have $\lambda^*(\boldsymbol{s})(b_i - \beta_i^*(\boldsymbol{s})) = \lambda^*(\boldsymbol{s}) \sum_{j \in [m]} p_j(\boldsymbol{s}) x_{ij}^*(\boldsymbol{s})$:

$$u_i(\boldsymbol{x}_i^*(\boldsymbol{s}); \boldsymbol{t}_i) - \lambda^*(\boldsymbol{s})(b_i - \beta_i^*(\boldsymbol{s})) = 0 \tag{85}$$

$$\lambda^*(\boldsymbol{s}) = \frac{u_i(\boldsymbol{x}_i^*(\boldsymbol{s}); \boldsymbol{t}_i)}{b_i - \beta_i^*(\boldsymbol{s})} \tag{86}$$

Combining the above conditions, with Equation (98), and adding to it Equation (82), and ensuring that the KKT primal feasibility conditions hold as well, we obtain the following necessary conditions that need to hold for all states $s \in \mathcal{S}$:

$$x^*_{ij}(s) > 0 \implies \frac{\frac{\partial u_i}{\partial x_{ij}}(x^*_i(s); t_i)}{p_j(s)} = \frac{u_i(x^*_i(s); t_i)}{b_i - \beta^*_i(s)} \qquad \forall j \in [m] \qquad (87)$$

$$\beta^*_i(s) > 0 \implies \frac{\partial \nu^*_i}{\partial b_i}(s) = \gamma \frac{\partial}{\partial \beta_i} \mathop{\mathbb{E}}_{(T', b', q')} [\nu^*_i(T', b' + \beta^*_i(s), q')] \qquad (88)$$

If additionally the transition probability function $p$ is CSD concave in $\beta_i$, then $\nu_i$ is concave and the utility maximization problem is concave, which in turn implies that the above conditions are also sufficient.

$\square$

**Theorem 4.1.** *A stochastic Fisher market with savings $\mathcal{F}$ in which $\mathcal{U}$ is a set of continuous and homogeneous utility functions and the transition function is continuous in $\beta_i$ has at least one recCE. Additionally, the recSE that solves the following Bellman equation corresponds to a recCE of $\mathcal{F}$:*

$$v(s) = \min_{p \in \mathbb{R}^m_+} \max_{(X, \beta) \in \mathbb{R}^{n \times (m+1)}_+ : Xp + \beta \leq b} \sum_{j \in [m]} q_j p_j + \sum_{i \in [n]} (b_i - \beta_i) \log(u_i(x_i, t_i))$$
$$+ \gamma \mathop{\mathbb{E}}_{(T', b', q') \sim p(\cdot | s, \beta)} [v(T', b' + \beta, q')] \qquad (13)$$

*Proof of Theorem 4.1.* Fix a buyer $i \in [n]$. Suppose that $v^*$ solves the following Stochastic Stackelberg game:

$$v(s) = \min_{p \in \mathbb{R}^m_+} \max_{(X, \beta) \in \mathbb{R}^{n \times (m+1)}_+ : Xp + \beta \leq b} \sum_{j \in [m]} q_j p_j + \sum_{i \in [n]} (b_i - \beta_i) \log(u_i(x_i, t_i))$$
$$+ \gamma \mathop{\mathbb{E}}_{(T', b', q') \sim p(\cdot | s, \beta)} [v(T', b' + \beta, q')] \qquad (89)$$

Define the Lagrangian associated with the following optimization problem:

$$\min_{p \in \mathbb{R}^m_+} \max_{(X, \beta) \in \mathbb{R}^{n \times (m+1)}_+ : Xp + \beta \leq b} \sum_{j \in [m]} q_j p_j + \sum_{i \in [n]} (b_i - \beta_i) \log(u_i(x_i, t_i))$$
$$+ \gamma \mathop{\mathbb{E}}_{(T', b', q') \sim p(\cdot | s, \beta)} [v^*(T', b' + \beta, q')] \qquad (90)$$

as follows:

$$\mathcal{L}(p, X, \beta, \lambda) = \sum_{j \in [m]} q_j p_j + \sum_{i \in [n]} (b_i - \beta_i) \log(u_i(x_i, t_i)) + \gamma \mathop{\mathbb{E}}_{(s') \sim p(\cdot | s, \beta)} [v(T', b' + \beta, q')]$$
$$+ \sum_{j \in [m]} \lambda_i (b_i - x_i \cdot p + \beta_i) \ . \qquad (91)$$

By Theorem 3.1, the necessary optimality conditions for the stochastic Stackelberg game are that for all states $s \in \mathcal{S}$ there exists $\mu^*(s) \in \mathbb{R}^{n \times (m+1)}$, and $\nu^*(s) \in \mathbb{R}^{n \times (m+1)}_+$ associated with the non-negativity constraints for $(X(s), \beta(s))$, and $p$ respectively, and $\lambda^*(s) \in \mathbb{R}^m_+$ associated with the spending constraint for $(X(s), \beta(s))$ such that:

$$q_j - \sum_{i \in [n]} \lambda^*_i(s) x^*_{ij}(s) - \nu^*_j(s) \doteq 0 \qquad \forall j \in [m]$$
$$(92)$$

$$\frac{b_i - \beta^*_i(s)}{u_i(x^*_i(s))} \frac{\partial u_i}{\partial x_{ij}}(x^*_i(s); t_i) - \lambda^*_i(s) p_j(s) + \mu^*_{ij}(s) \doteq 0 \qquad \forall i \in [n], j \in [m]$$
$$(93)$$

$$- \log(u_i(x^*_i(s))) + \gamma \frac{\partial}{\partial \beta_i} \mathop{\mathbb{E}}_{(T', b', q')} [v(T', b' + \beta^*(s), q')] - \lambda^*_i(s) + \mu^*_{i(m+1)}(s) \doteq 0 \quad \forall i \in [n]$$
$$(94)$$

Note that by Theorem 3.1, we also have $\mu^*_{i(m+1)}(\boldsymbol{s})\beta^*_i(\boldsymbol{s}) = \mu^*_{i+1)}(\boldsymbol{s})x^*_{ij}(\boldsymbol{s}) = 0$ which gives us:

$$p_j(\boldsymbol{s}) > 0 \implies q_j - \sum_{i \in [n]} \lambda^*_i(\boldsymbol{s})x^*_{ij}(\boldsymbol{s}) = 0 \qquad \forall j \in [m] \quad (95)$$

$$x^*_{ij}(\boldsymbol{s}) > 0 \implies \frac{b_i - \beta^*_i(\boldsymbol{s})}{u_i(\boldsymbol{x}^*_i(\boldsymbol{s}))}\frac{\partial u_i}{\partial x_{ij}}(\boldsymbol{x}^*_i(\boldsymbol{s})) - \lambda^*_i(\boldsymbol{s})p_j(\boldsymbol{s}) = 0 \qquad \forall i \in [n], j \in [m] \quad (96)$$

$$\beta^*_i(\boldsymbol{s}) > 0 \implies -\log\left(u_i(\boldsymbol{x}^*_i(\boldsymbol{s}); \boldsymbol{t}_i)\right) + \gamma\frac{\partial}{\partial \beta_i} \mathop{\mathbb{E}}_{(\boldsymbol{T}', \boldsymbol{b}', \boldsymbol{q}')}\left[v(\boldsymbol{T}', \boldsymbol{b}' + \boldsymbol{\beta}^*(\boldsymbol{s}), \boldsymbol{q}')\right]$$
$$- \lambda^*_i(\boldsymbol{s}) + \mu^*_{i(m+1)}(\boldsymbol{s}) = 0 \qquad \forall i \in [n] \quad (97)$$

Re-organizing expressions, we obtain:

$$p_j(\boldsymbol{s}) > 0 \implies q_j = \sum_{i \in [n]} \lambda^*_i(\boldsymbol{s})x^*_{ij}(\boldsymbol{s}) \qquad \forall j \in [m] \quad (98)$$

$$x^*_{ij}(\boldsymbol{s}) > 0 \implies \frac{u_i(\boldsymbol{x}^*_i(\boldsymbol{s}))}{b_i - \beta^*_i(\boldsymbol{s})}\lambda^*_i(\boldsymbol{s}) = \frac{\frac{\partial u_i}{\partial x_{ij}}(\boldsymbol{x}^*_i(\boldsymbol{s}))}{p_j(\boldsymbol{s})} \qquad \forall i \in [n], j \in [m] \quad (99)$$

$$\beta^*_i(\boldsymbol{s}) > 0 \implies -\log\left(u_i(\boldsymbol{x}^*_i(\boldsymbol{s}))\right) + \gamma\frac{\partial}{\partial \beta_i} \mathop{\mathbb{E}}_{(\boldsymbol{T}', \boldsymbol{b}', \boldsymbol{q}')}\left[v(\boldsymbol{T}', \boldsymbol{b}' + \boldsymbol{\beta}^*(\boldsymbol{s}), \boldsymbol{q}')\right]$$
$$- \lambda^*_i(\boldsymbol{s}) = 0 \qquad \forall i \in [n] \quad (100)$$

Using the envelope theorem, we can compute $\frac{\partial v}{\partial b_i}$ as follows:

$$\frac{\partial v^*}{\partial b_i}(\boldsymbol{s}) = \log\left(u_i(\boldsymbol{x}^*_i(\boldsymbol{s}); \boldsymbol{t}_i)\right) + \lambda^*_i(\boldsymbol{s}) \qquad (101)$$

Once again note that $\frac{\partial v}{\partial b_i}$ is well-defined by Equation (99).

Re-organizing expressions, we get:

$$\lambda^*_i(\boldsymbol{s}) = \frac{\partial v}{\partial b_i}(\boldsymbol{s}) - \log\left(u_i(\boldsymbol{x}^*_i(\boldsymbol{s}); \boldsymbol{t}_i)\right) \qquad (102)$$

Combining Equation (102) and Equation (100), we obtain:

$$\beta^*_i(\boldsymbol{s}) > 0 \implies -\log\left(u_i(\boldsymbol{x}^*_i(\boldsymbol{s}); \boldsymbol{t}_i)\right) + \gamma\frac{\partial}{\partial \beta_i} \mathop{\mathbb{E}}_{(\boldsymbol{T}', \boldsymbol{b}', \boldsymbol{q}')}\left[v(\boldsymbol{T}', \boldsymbol{b}' + \boldsymbol{\beta}^*(\boldsymbol{s}), \boldsymbol{q}')\right]$$
$$- \frac{\partial v}{\partial b_i}(\boldsymbol{s}) + \log\left(u_i(\boldsymbol{x}^*_i(\boldsymbol{b}); \boldsymbol{t}_i)\right) = 0 \qquad \forall i \in [n] \quad (103)$$

$$\beta^*_i(\boldsymbol{s}) > 0 \implies \gamma\frac{\partial}{\partial \beta_i} \mathop{\mathbb{E}}_{(\boldsymbol{T}', \boldsymbol{b}', \boldsymbol{q}')}\left[v(\boldsymbol{T}', \boldsymbol{b}' + \boldsymbol{\beta}^*(\boldsymbol{s}), \boldsymbol{q}')\right] - \frac{\partial v}{\partial b_i}(\boldsymbol{s}) = 0 \qquad \forall i \in [n] \quad (104)$$

$$\beta^*_i(\boldsymbol{s}) > 0 \implies \frac{\partial v}{\partial b_i}(\boldsymbol{s}) = \gamma\frac{\partial}{\partial \beta_i} \mathop{\mathbb{E}}_{(\boldsymbol{T}', \boldsymbol{b}', \boldsymbol{q}')}\left[v(\boldsymbol{T}', \boldsymbol{b}' + \boldsymbol{\beta}^*(\boldsymbol{s}), \boldsymbol{q}')\right] \qquad \forall i \in [n] \quad (105)$$

Going back to Equation (93), multiplying both sides by $x^*_{ij}(\boldsymbol{s})$ and summing up across all $j \in [m]$, we get:

$$\sum_{j \in [m]} \frac{b_i - \beta^*_i(\boldsymbol{s})}{u_i(\boldsymbol{x}^*_i(\boldsymbol{s}); \boldsymbol{t}_i)} \sum_{j \in [m]} x^*_{ij}(\boldsymbol{s})\frac{\partial u_i}{\partial x_{ij}}(\boldsymbol{x}^*_i(\boldsymbol{s})) - \lambda^*_i(\boldsymbol{s})\sum_{j \in [m]} p_j(\boldsymbol{s})x^*_{ij}(\boldsymbol{s}) = 0 \qquad (106)$$

$$\frac{b_i - \beta^*_i(\boldsymbol{s})}{u_i(\boldsymbol{x}^*_i(\boldsymbol{s}); \boldsymbol{t}_i)}u_i(\boldsymbol{x}^*_i(\boldsymbol{s}); \boldsymbol{t}_i) - \lambda^*_i(\boldsymbol{s})\sum_{j \in [m]} p_j(\boldsymbol{s})x^*_{ij}(\boldsymbol{s}) = 0 \quad \text{(Euler's Theorem)} \qquad (107)$$

$$b_i - \beta^*_i(\boldsymbol{s}) - \lambda^*_i(\boldsymbol{s})\sum_{j \in [m]} p_j(\boldsymbol{s})x^*_{ij}(\boldsymbol{s}) = 0 \qquad (108)$$

By Theorem 3.1, we have that $\lambda_i^*(\boldsymbol{s})\left(b_i - \sum_{j\in[m]} p_j(\boldsymbol{s})x_{ij}^*(\boldsymbol{s}) - \beta_i^*(\boldsymbol{s})\right) = 0$, which gives us:

$$b_i - \beta_i^*(\boldsymbol{s}) - \lambda_i^*(\boldsymbol{s})(b_i - \beta_i^*(\boldsymbol{s})) = 0 \tag{109}$$

$$\lambda_i^*(\boldsymbol{s}) = 1 \tag{110}$$

Combining the above with Equations (98) to (100) we obtain:

$$p_j(\boldsymbol{s}) > 0 \implies q_j = \sum_{i\in[n]} x_{ij}^*(\boldsymbol{s}) \qquad\qquad \forall j \in [m] \tag{111}$$

$$x_{ij}^*(\boldsymbol{s}) > 0 \implies \frac{u_i(\boldsymbol{x}_i^*(\boldsymbol{s}))}{b_i - \beta_i^*(\boldsymbol{s})} = \frac{\frac{\partial u_i}{\partial x_{ij}}(\boldsymbol{x}_i^*(\boldsymbol{s}); \boldsymbol{t}_i)}{p_j(\boldsymbol{s})} \qquad\qquad \forall i \in [n], j \in [m] \tag{112}$$

$$\beta_i^*(\boldsymbol{s}) > 0 \implies \frac{\partial v}{\partial b_i}(\boldsymbol{b}) = \gamma \frac{\partial}{\partial \beta_i} \mathop{\mathbb{E}}_{(\boldsymbol{T'},\boldsymbol{b'},\boldsymbol{q'})}[v(\boldsymbol{T'}, \boldsymbol{b'} + \boldsymbol{\beta}^*(\boldsymbol{s}), \boldsymbol{q'})] \quad \forall i \in [n] \tag{113}$$

Since the utility functions are non-satiated, and by the second equation, the buyers are utility maximizing at state $\boldsymbol{s}$ over all allocations, we must also have that Walras' law holds, i.e., $\boldsymbol{p} \cdot \left(\boldsymbol{q} - \sum_{i\in[n]} \boldsymbol{x}_i\right) - \sum_{i\in[n]} \beta_i = 0$. Walras' law combined with the first equation above then imply the second condition of a recursive competitive equilibrium. Finally, by Lemma D.1, the last two equations imply the first condition of recursive competitive equilibrium.

$\square$

# E   Experiment Details

## E.1   Stochastic Fisher market without interest rates

For the without interest rates setup, we initialized a stochastic Fisher market with $n = 2$ buyers and $m = 2$ goods. To simplify the analysis, we assumed deterministic transitions such that the buyers get a constant new budgets of 9.5 at each time period, and their types/valuations as well as the supply of goods does not change at each state, i.e., the type/valuation space and supply space has cardinality 1. This reduced the market to a deterministic repeated market setting in which the amount of budget saved by the buyers differentiates different states of the market. To initialize the state space of the market, we first fixed a range of $[10, 50]^m$ for the buyers' valuations and drew for all buyers $i \in [n]$ valuations $\boldsymbol{\theta}_i$ from that range uniformly at random at the beginning of the experiment. (We scaled the valuations differently for different markets to ensure positive utilities though.) We have assumed the supply of goods is $\boldsymbol{1}_m$ and that the budget space was $[9, 10]^n$. This means that our state space for our experiments was $\mathcal{S} = \{(\boldsymbol{\theta}_1, \boldsymbol{\theta}_2)\} \times \{\boldsymbol{1}_m\} \times [9, 10]^n$. We note that although the assumption that buyers valuations/type space has cardinality one does simplify the problem, the supply of the goods being $\boldsymbol{1}$ at each state is wlog because goods are divisible and the allocation of goods to buyers at each state can then be interpreted as the percentage of a particular good allocated to a buyer. We assumed initial budgets of $\boldsymbol{b}^{(0)} = 10_n$ for buyers.

Since the state space is continuous, the value function has continuous domain in the stochastic Fisher market setting. As a result, we had to use fitted variant of value iteration. In particular, we assumed that the value function had a linear form at each state such that $v(\boldsymbol{T}, \boldsymbol{b}, \boldsymbol{q}; \boldsymbol{a}, c) = \boldsymbol{a}^T \boldsymbol{b} + c$ for some parameters $\boldsymbol{a} \in \mathbb{R}^n, c \in \mathbb{R}$, and we tried to approximate the value function at the next step of value iteration by using linear regression. That is, at each value iteration step, we uniformly sampled 25 budget vectors from the range $[9, 10]^n$. Next, for each sampled budget $\boldsymbol{b}$, we solved the min-max step given that budget as a state. This process gave us (budget, value) pairs on which we ran linear regression to approximate the value function at the next iterate.

To solve the generalized min-max operator at each step of value iteration, we used the **nested gradient descent ascent (GDA)** [6] (Algorithm 3) along with JAX gradients, which is not guaranteed to converge to a global optimum since the min-max Stackelberg game for stochastic Fisher markets is convex-non-concave. Then, we have run value iteration for 30 iterations. We ran nested GDA with learning rates $\eta_{\boldsymbol{X}} = 1.4, \eta_{\boldsymbol{p}} = 1.5 \times 10^{-2}$ for linear, $\eta_{\boldsymbol{X}} = 1.5, \eta_{\boldsymbol{p}} = 6.5 \times 10^{-4}$ for leontief, and $\eta_{\boldsymbol{X}} = 1.4, \eta_{\boldsymbol{p}} = 5 \times 10^{-3}$ for Cobb-Douglas. The outer loop of nested GDA was run for $T_{\boldsymbol{p}} = 60$

iterations, while its inner loop was run for $T_X = 100$ iterations, and break from the nested GDA if we obtain an excess demand with norm lower than 0.01. We depict the trajectory of the average value of the value function at each iteration of value iteration under nested GDA in Figure 1.

### E.2 Stochastic Fisher market with interest rates

For the with interest rates setup, we initialized a stochastic Fisher market with $n = 5$ buyers and $m = 5$ goods. This time, we implemented a stochastic transitions. Though buyers still get a constant new budgets of 9.5 at each time step, and their types/valuations as well as the supply of goods does not change at each state, their savings from last time step may decrease or increase according to some probabilistic interest rates. In specific, at each time step, we consider five interest rates $\{0.9, 1.0, 1.1, 1.2, 1.5\}$, each with probability 0.2. Thus, we have a stochastic market setting in which the amount of budget possessed by the buyers at the beginning of each time step differentiates different states of the market. To initialize the state space of the market, we first fixed a range of $[0, 1]^m$ for the buyers' valuations and drew for all buyers $i \in [n]$ valuations $\boldsymbol{\theta}_i$ from that range uniformly at random at the beginning of the experiment. (We scaled the valuations differently for different markets to ensure positive utilities though.) We have assumed the supply of goods is $\mathbf{1}_m$ and that the budget space was $[9, 10]^n$. This means that our state space for our experiments was $\mathcal{S} = \{(\boldsymbol{\theta}_1, \boldsymbol{\theta}_2, \boldsymbol{\theta}_3, \boldsymbol{\theta}_4, \boldsymbol{\theta}_5)\} \times \{\mathbf{1}_m\} \times [9, 10]^n$. We note that although the assumption that buyers valuations/type space has cardinality one does simplify the problem, the supply of the goods being $\mathbf{1}$ at each state is wlog because goods are divisible and the allocation of goods to buyers at each state can then be interpreted as the percentage of a particular good allocated to a buyer. We assumed initial budgets of $\boldsymbol{b}^{(0)} = 10_n$ for buyers.

Since the state space is continuous, the value function has continuous domain in the stochastic Fisher market setting. As a result, we had to use fitted variant of value iteration. In particular, we assumed that the value function had a linear form at each state such that $v(\boldsymbol{T}, \boldsymbol{b}, \boldsymbol{q}; \boldsymbol{a}, c) = \boldsymbol{a}^T \boldsymbol{b} + c$ for some parameters $\boldsymbol{a} \in \mathbb{R}^n, c \in \mathbb{R}$, and we tried to approximate the value function at the next step of value iteration by using linear regression. That is, at each value iteration step, we uniformly sampled 25 budget vectors from the range $[9, 10]^n$. Next, for each sampled budget $\boldsymbol{b}$, we solved the min-max step given that budget as a state. This process gave us (budget, value) pairs on which we ran linear regression to approximate the value function at the next iterate.

To solve the generalized min-max operator at each step of value iteration, we used the **nested gradient descent ascent (GDA)** [6] (Algorithm 3) along with JAX gradients, which is not guaranteed to converge to a global optimum since the min-max Stackelberg game for stochastic Fisher markets is convex-non-concave. Then, we have run value iteration for 30 iterations. We ran nested GDA with learning rates $\eta_X = 1.7, \eta_p = 2 \times 10^{-2}$ for linear, $\eta_X = 2, \eta_p = 5 \times 10^{-5}$ for leontief, and $\eta_X = 1.8, \eta_p = 2.5 \times 10^{-2}$ for Cobb-Douglas. The outer loop of nested GDA was run for $T_p = 60$ iterations, while its inner loop was run for $T_X = 100$ iterations, and break from the nested GDA if we obtain an excess demand with norm lower than 0.01. We depict the trajectory of the average value of the value function at each iteration of value iteration under nested GDA in Figure 2.

### E.3 Other Details

**Programming Languages, Packages, and Licensing**   We ran our experiments in Python 3.7 [72], using NumPy [73], CVXPY [74], and JAX [63]. Figure 1 and Figure 2 were graphed using Matplotlib [75].

Python software and documentation are licensed under the PSF License Agreement. Numpy is distributed under a liberal BSD license. Matplotlib only uses BSD compatible code, and its license is based on the PSF license. CVXPY is licensed under an APACHE license.

**Implementation Details**   In our execution of Algorithm 3, in order to project each allocation computed onto the budget set of the consumers, i.e., $\{\boldsymbol{X} \in \mathbb{R}_+^{n \times m} \mid \boldsymbol{X}\boldsymbol{p} \leq \boldsymbol{b}\}$, we used the CVXPY with MOSEK solver with warm start option, a feature that enables the solver to exploit work from previous solves.

**Computational Resources**   Our experiments were run on Google Colab with 12.68GB RAM, and took about 8 hours to run the Stochastic Fisher market without interest rates experiment (for each

utility function class) and about 8.5 hours to run the Stochastic Fisher market with interest rates experiment (for each utility function class). Only CPU resources were used.

**Code Repository**   The data our experiments generated, and the code used to produce our visualizations, can be found in our code repository (https://github.com/Sadie-Zhao/Zero-Sum-Stochastic-Stackelberg-Games-NeurIPS).