# OpenReview forum: "Zero-Sum Stochastic Stackelberg Games"
_NeurIPS.cc/2022/Conference — NeurIPS 2022 Accept_

### Official Review · Reviewer_CYu7 · 2022-07-01

**Rating:** 7
**Confidence:** 4
**Soundness:** 4 excellent
**Presentation:** 3 good
**Contribution:** 3 good

**Summary:**

The paper introduces a novel variant of the stochastic game model, where the available strategies of one player may be constrained by strategy choice of the other player. The paper establishes the existence of a corresponding solution concept and shows that a variant of value iteration approaches it. The paper shows the relevance of the proposed model by using it to model a realistic variant of Fisher market. Finally, the paper demonstrates that the algorithm converges very close to the solution in some games and a little further in more complex games.

**Questions:**

What is the intuitive reason why the solution does not depend on the initial distribution?


**Limitations:**

The limitations are discussed and I do not think there is a significant societal impact.

**Strengths And Weaknesses:**

I enjoyed reading the paper. The paper is not very well motivated in the introduction, but the separate Section 4 already nicely demonstrates the usefulness of the model. The results in the paper are not very surprising, but still important to establish for a new game model. The content is quite technical, but notation is intuitive and definitions sufficiently thorough to make the paper reasonably easy to follow.

I have only smaller suggestions for improvements:

I understand the motivation by the Stackelberg setting.and I like the name of the model. However, referring to the distinction from the standard model as static vs. dynamic is very confusing for me. Stochastic games are fundamentally dynamic and referring to the games as static/dynamic based on what happens in individual states makes the claims, e.g. in the abstract, sound just wrong.

Lines 49-51: It was completely unclear to me how the mentioned games could benefit from being modeled as Stackelberg games.

Line 65-66: The line of 22 references without giving any context is useless and makes it look like an artificial inflation of the number of citations. What is the reader supposed to learn from that? Since this paper is about the Slackelberg setting, these papers do not seem relevant at all.

Lines 69-78: Similar as above, I am not sure what is the relation of these papers to the current work. Few lines of explanation would be very appreciated.

Line 243: I guess you mean equations (1)-(5).

I would appreciate a plain text explanation of equation (6) to make it faster to understand.

---

> ### Author Response · Authors · 2022-08-01
> **Response to reviewer CYu7**
>
> Thank you for your review and suggestions!
>
> We realize from your comments that we have not emphasized non-economic applications of our game model enough, so we will emphasize these applications to further motivate our model. In particular, as your review mentions, applications of zero-sum stochastic Stackelberg games include but are not limited to autonomous driving [1], reach-avoid problems in human-robot interaction [2], and robust optimization in stochastic environments [3]. All these problems employ the game model we study, but do not provide any theoretical guarantees; our work provides a theoretical framework in which to analyze the applications studied in these papers. Indeed, the initial motivation for our work came from conversations with experts in human-robot interaction, who suggested that our game model was the basic framework used in popular reach-avoid problems. Here is a high-level explanation of how zero-sum stochastic Stackelberg games model the reach-avoid problem:
>
> The reach-avoid problem consists of two agents: the first of these agents, the defender, is trying to reach a goal location without being caught by the other agent, the attacker, who is trying to keep the defender away from the goal location. This problem can be modeled as a zero-sum stochastic Stackelberg game in which the attacker is the leader and the defender is the follower. The follower’s reward is higher when it is closer to the goal location, while the leader earns the negative of this reward. The constraints $\mathbf{g}$ ensure that the defender always remains a given distance from the attacker, while the rewards ensure that the defender is incentivized to reach the goal location.
>
> Thank you for mentioning that the distinction we make between dynamic and static games can be confusing! We will do our best to address this point.
>
> Regarding your suggestions:
>
> Lines 49-51: We will be sure to describe how zero-sum stochastic Stackelberg games model these additional games, just as we explain the reach-avoid problem above.
>
> Line 65-66: We will prune most of these references.
>
> Lines 69-78: The models in the papers references in lines 69-78 differ from ours in various way. Moreover, and more notable, none of those papers provide polynomial-time computation guarantees for the algorithms they consider, while we do. Similarly, none of the references brought to our attention by reviewer nfFs provide algorithms with polynomial-time computation guarantees. As such, ours is the first polynomial-time algorithm for computing Stackelberg equilibria in any class of stochastic Stackelberg games.
>
> Line 243: Thank you for noticing this! We will fix the referencing error.
>
> Equation 6: We will insert an English description of equation 6. At a high level, the equation states that the policies that solve the zero-sum stochastic Stackelberg game, in which sellers pick prices to minimize the immediate economic surplus at each state and buyers pick allocations and savings to maximize the immediate economic surplus at each state constrained by their spending and saving constraints, correspond to a recCE, which is characterized by a value function/policies that maximize the long-term economic surplus at each state.
>
> Regarding your question:
>
> In our problem formulation, we assume that the model of the environment is known; thus, we consider a planning problem, not a learning problem. As such, value iteration finds the optimal value at all states, which is independent of the initial state distribution, just as it is in an MDP [4].
>
> [1] Jaime F Fisac et al. “Reach-avoid problems with time-varying dynamics, targets and constraints”. In: Proceedings of the 18th international conference on hybrid systems: computation and control. 2015, pp. 11–20.
>
> [2] Somil Bansal et al. “Hamilton-jacobi reachability: A brief overview and recent advances”. In:
> 2017 IEEE 56th Annual Conference on Decision and Control (CDC). IEEE. 2017, pp. 2242–358
> 2253
>
> [3] Dimitris Bertsimas, David B Brown, and Constantine Caramanis. “Theory and applications of robust optimization”. In: SIAM review 53.3 (2011), pp. 464–501
>
> [4] Bellman, Richard. "The theory of dynamic programming." Bulletin of the American Mathematical Society 60.6 (1954): 503-515.

---

> > ### Comment · Reviewer_CYu7 · 2022-08-05
> > **No change in opinion**
> >
> > I have read the rebuttal and the other reviews. While I generally agree with the limitations raised by the other reviewers, I do not see then as preventing the publication of the paper, which I still believe is sufficiently novel and interesting.

---

### Official Review · Reviewer_j5ee · 2022-07-11

**Rating:** 5
**Confidence:** 3
**Soundness:** 3 good
**Presentation:** 2 fair
**Contribution:** 2 fair

**Summary:**

This paper studies zero-sum Stackelberg games. First, this paper proves the existence of the recursive Stackelberg equilibrium. Furthermore, the authors provide necessary and sufficient conditions for identifying the Stackelberg equilibrium and show that a recursive Stackelberg equilibrium can be computed in polynomial time via value iteration. Finally, the authors show that the stochastic Fisher market can be modeled by the stochastic Stackelberg games.

**Questions:**

1. 	It seems that some results (e.g. value iteration) are straightforward. I hope the authors point out the non-trivial part if I miss something. I also encourage the authors to highlight the technique novelties in the paper.
2. 	This paper only considers planning problems where the model is known. Recently, some papers considers learning problems in Stackelberg games [1] [2]. Can these methods (with some modifications) be applied to finding recursive Stackelberg equilibria in online zero-sum Stackelberg games? I hope the authors make some comments on this.

[1] Y. Bai, C. Jin, H. Wang, and C. Xiong. Sample-efficient learning of Stackelberg equilibria in general-sum games. In NeurIPS, 2021.

[2] H. Zhong, Z. Yang, Z. Wang, and M. I. Jordan. Can reinforcement learning find Stackelberg-Nash equilibria in general-sum Markov games with myopic followers? 2021. arXiv:2112.13521.



**Limitations:**

Yes

**Strengths And Weaknesses:**

This paper studies the zero-sum Stackelberg game and provides a complete set of results for this problem. The authors also show that the zero-sum Stackelberg game can model the Fisher market. This is interesting. But I still have some concerns (see the questions below).

---

> ### Author Response · Authors · 2022-08-01
> **Response to reviewer j5ee**
>
>
> Thank you for your review and questions!
>
> To answer your first questions, our paper includes four technical contributions, which together make our results a non-trivial extension of Shapley’s results on zero-sum stochastic games. Our first technical contribution is to show that the generalized min-max operator is a non-expansive operator, i.e., it is 1-Lipschitz w.r.t. the sup-norm (Lemma 2.3). This result is quite surprising since constrained parametric optimization problems with parameterized constraints are not in general Lipschitz in their parameters even when the objective to be optimized is Lipschitz. That is, $\max_{\mathbf{y} \in \mathcal{Y}: \mathbf{g}(\mathbf{x}, \mathbf{y}) \geq 0} f(\mathbf{x}, \mathbf{y})$ is not guaranteed to be Lipschitz even when $f$ and $\mathbf{g}$ are Lipschitz. This is in stark contrast to the result that holds in Shapley’s setting, i.e., when the constraints are not parameterized, namely that $\max_{\mathbf{y} \in \mathcal{Y}} f(\mathbf{x}, \mathbf{y})$ is Lispchitz when $f$ is Lipschitz. Our result is even more surprising since Lemma 2.3 holds without even assuming Lipschitzness of the objective or constraints.
>
> Our second technical contribution is to show that the value function can be expressed recursively, and is thus amenable to dynamic programming: if the players optimize their policies locally/greedily at each state wrt their long-term values, the ensuing values and policies will be globally optimal. This result may be surprising to some, since it says that a zero-sum stochastic Stackelberg game can be seen both as a sequential game in a global sense (i.e. leader picks a complete policy before first, then the follower best-responds to this policy with their own complete policy), and as a sequential game in a local sense (i.e., players play a Stackelberg game at each state in which the leader picks an action and the follower best-responds).
>
> Our third technical contribution is to provide a subdifferential generalization of the Benveniste-Scheinkman theorem, which allows one to subdifferentiate the optimal value function associated with an MDP with parameterized rewards and action spaces. We use this theorem to prove that stochastic Fisher markets can be solved as zero-sum stochastic Stackelberg games, but this theorem may have other applications as well, such as proving that a given MDP satisfies certain optimality conditions.
>
> This brings us to our fourth and final technical contribution. We provide the first mathematical characterization of recCE in a stochastic market model using our subdifferential generalization of the Benveniste-Scheinkman theorem. Our proof technique is to derive the optimality conditions for the zero-sum stochastic Stackelberg game associated with a stochastic Fisher market, thereby proving that optimal policies for the game correspond to a recCE of the stochastic Fisher market. To the best of our knowledge, this is the first result of its kind. Multiple others have modeled markets as stochastic games [2,3,4,5], without characterizing their solutions mathematically. Our proof technique may be useful in this regard.
>
> We will be sure to include some of the above discussion to clarify our technical contributions.
>
> Regarding your second question, it is not immediately obvious to us that the methods provided in the papers you referenced [6,7] can be used to learn stationary Stackelberg policies in model-free zero-sum stochastic Stackelberg games. In the Bandit-RL model [6], the leader only picks an action and not a policy; additionally the authors consider non-discounted payoffs and a finite horizon. In [7], the authors consider a finite horizon model, where the action of the leader at each state does not constrain the actions available to the followers. However, as we mentioned to reviewer nfFs, we believe that minimax-Q learning can be generalized to our setting, and future work could prove sample complexity bounds for such an algorithm. All of that said, both of the papers you mention [6,7] are relevant literature in this direction, so we will be sure to add them to provide future directions for other researchers.
>
> (You can find the references included in the following response)

---

> > ### Author Response · Authors · 2022-08-01
> > **References for Response**
> >
> > References are included below due to lack of space
> >
> > [1] Shapley, Lloyd S. "Stochastic games." Proceedings of the national academy of sciences 39.10 (1953): 1095-1100.
> >
> > [2] Zheng, Stephan, et al. "The AI Economist: Optimal Economic Policy Design via Two-level Deep Reinforcement Learning." arXiv preprint arXiv:2108.02755 (2021).
> >
> > [3] Zheng, Stephan, et al. "The ai economist: Improving equality and productivity with ai-driven tax policies." arXiv preprint arXiv:2004.13332 (2020).
> >
> > [4] Curry, Michael, et al. "Finding General Equilibria in Many-Agent Economic Simulations Using Deep Reinforcement Learning." arXiv preprint arXiv:2201.01163 (2022).
> >
> > [5] Hill, Edward, Marco Bardoscia, and Arthur Turrell. "Solving heterogeneous general equilibrium economic models with deep reinforcement learning." arXiv preprint arXiv:2103.16977 (2021).
> >
> > [6] Y. Bai, C. Jin, H. Wang, and C. Xiong. Sample-efficient learning of Stackelberg equilibria in general-sum games. In NeurIPS, 2021.
> >
> > [7] H. Zhong, Z. Yang, Z. Wang, and M. I. Jordan. Can reinforcement learning find Stackelberg-Nash equilibria in general-sum Markov games with myopic followers? 2021. arXiv:2112.13521.
> >
> > [8] Littman, M. L. (1994). Markov games as a framework for multi-agent reinforcement learning. In Machine learning proceedings 1994 (pp. 157-163). Morgan Kaufmann.

---

### Official Review · Reviewer_nfFs · 2022-07-12

**Rating:** 6
**Confidence:** 3
**Soundness:** 3 good
**Presentation:** 2 fair
**Contribution:** 2 fair

**Summary:**

The authors analyze the zero-sum stochastic Stackelberg game. They proved the existence of an optimal solution in zero-sum games and showed how to compute a recursive Stackelberg equilibrium via value iteration. Then they provide some applications of stochastic Stackelberg games.

**Questions:**

1. Could Lemma 2.1 be written only from the "equivalently" part? The first part seems redundant.

2. It would be useful to add citations on how to solve the min-max constrained problem defined by the contraction operator.

3. How does the proposed method differ from the min-max algorithm from [1]?

4. Remark 2.6 said that the payoff function can also be non-convex-non-concave. How can we solve the optimization problem given by the contraction operator if the payoff function is non-convex-non-concave?

5. How can the algorithm be independent of the initial-state distribution? If we start from a state $s$ which is disconnected from the other states we are achieving a different recSE... or am I missing something (maybe connected to Assumption 1)?

6. I find section 3 quite interesting but not well commented. I suggest explaining better what the implications of the proposed theorem are.

7. Although less interesting, it would be useful to add experiments with finite state-action spaces, where the authors can use the value-iteration method. Is it possible to use [2] as a baseline?

8. Are there other applications of the game considered?

[1] Littman, M. L. (1994). Markov games as a framework for multi-agent reinforcement learning. In Machine learning proceedings 1994 (pp. 157-163). Morgan Kaufmann.

[2] Fiez, T., Chasnov, B., & Ratliff, L. (2020, November). Implicit learning dynamics in stackelberg games: Equilibria characterization, convergence analysis, and empirical study. In International Conference on Machine Learning (pp. 3133-3144). PMLR.

**Limitations:**

Since there is space the authors could add a section analyzing the limitation and the potential negative societal impact of their work. However, since the work is more theoretical, I don't think it has an immediate negative societal impact.

**Strengths And Weaknesses:**

Strengths
1. The paper studies an interesting problem: a leader-follower game where the follower's actions depend on the leader's actions.

2. The authors propose (as far as I know) a novel solution concept recSE and study when this equilibrium exists in stochastic games.

3. The authors propose a value-iteration algorithm to converge to the equilibrium solution.


Weaknesses
1. The related work does not mention the works on Stochastic Stackelberg Games and Leader-follower stochastic Stackelberg games as [1,2,3,4,5]. What is the relation between the considered setting and these other ones? Maybe the name stochastic Stackelberg game is not the most appropriate.

2. Line 129, 140, 264 have formatting problems. $\alpha$ is an unusual way to bound the rewards I suggest putting 1 or $R_\max$.

3. I have some questions which I listed below.

[1] Vorobeychik, Y., & Singh, S. (2012). Computing stackelberg equilibria in discounted stochastic games. In Proceedings of the AAAI Conference on Artificial Intelligence (Vol. 26, No. 1, pp. 1478-1484).

[2] Chang, Yanling, Alan L. Erera, and Chelsea C. White. "A leader–follower partially observed, multiobjective Markov game." Annals of Operations Research 235.1 (2015): 103-128.

[3] Sengupta, S., & Kambhampati, S. (2020). Multi-agent reinforcement learning in bayesian stackelberg markov games for adaptive moving target defense. arXiv preprint arXiv:2007.10457.

[4] Ramponi, G., & Restelli, M. (2022, February). Learning in Markov Games: can we exploit a general-sum opponent?. In The 38th Conference on Uncertainty in Artificial Intelligence.

[5] Vu, Q. L., Alumbaugh, Z., Ching, R., Ding, Q., Mahajan, A., Chasnov, B., ... & Ratliff, L. J. (2022, March). Stackelberg Policy Gradient: Evaluating the Performance of Leaders and Followers. In ICLR 2022 Workshop on Gamification and Multiagent Solutions.

---

> ### Author Response · Authors · 2022-08-01
> **Response to Reviewer nfFs**
>
> Thank you for your review and pointing us to more relevant literature! We would like to clarify the relationship between our model and those presented in the papers you have referenced. First of all, our paper considers a class of stochastic Stackelberg games, namely zero-sum, for which, to our knowledge, we provide the first polynomial-time computation guarantees. The references you mention [1,2,3,4,5]  do not provide polynomial-time computation guarantees.
>
> In terms of models, the most important difference between ours and the ones in the papers you reference is that we consider a setting that generalizes other Stackelberg settings, in that the action chosen by the leader at each state can affect the actions that are available to the follower at that state. In all the papers you mention [1,2,3,4,5], the followers’ action sets are fixed a priori, and thus not impacted by the actions chosen by the leader.
>
> We can of course add citations to these other papers on Stackelberg games, and space permitting, outline the following differences between our models and theirs: Our paper as well as [1,4,5] consider a complete information setting, while [2,3] assume an incomplete information setting. While our model as well as that of [5] make no assumptions on the discreteness or continuity of the action set, papers [1,2,3,4] assume that the action spaces are discrete. While our model and that of [2,5] make no discreteness or continuity assumptions on the state space, [1,3,4] assume that the state space is discrete. Our paper as well as that of [2,3] assume an infinite game horizon, while [1,4,5] are finite horizon settings. Our paper as well as [1,2,3,5] assume discounted payoffs, while [4] assumes non-discounted payoffs.  Similar to [2,3 5], we restrict ourselves to stationary policies, while [1,4] consider non-stationary policies.
>
> We use the name Stackelberg games following the bilevel optimization literature [16], as our model is a special case of bilevel optimization.
>
> Regarding your questions:
>
> 1. Yes, Lemma 2.1 consists of two equivalent statements. The second part characterizes the Stackelberg equilibrium policies, while the first part characterizes the optimal action-value function at a Stackelberg equilibrium. We will be sure to clarify these points.
>
> 2. To the best of our knowledge, we have added all known references on solving min-max Stackelberg games [6,7]. We would be happy to include additional relevant references, should you know of any that we are missing.
>
> 3. Our game model is more general than that of [8], while our problem domain is less general. While we model two-player stochastic games in which the actions taken by the players determine both the payoffs *and the set of feasible actions*, [8] considers two-player stochastic games in which the players’ actions determine *only payoffs*. On the other hand, in our problem formulation, we assume that the model of the environment is known, while [8] does not. Future work could aim to provide a generalization of minimax-Q learning to zero-sum stochastic Stackelberg games. We believe that such a generalization should be possible, but this learning problem falls outside the scope of the present paper.
>
> 4. Remark 2.6 refers to the existence of a solution only, not to its computation. That said, although we cannot obtain polynomial-time computation guarantees for the generalized min-max operator in non-convex-non-concave settings, there exist many heuristic methods, however, that converge in practice to global optima for such problems (see for instance [9] or [10]). In our experiments we use such a heuristic, namely simulated annealing, to solve stochastic Fisher markets with savings, a problem which is convex-non-concave.
>
> 5. Once again, since we consider only the planning problem, value iteration finds the optimal value at all states, which is independent of the initial state distribution, just as it is in a MDP [11]. Thus, we do not face issues such as not being able to reach a particular state, which may occur in a learning problem. This issue would be an important one to consider in future work aimed at learning optimal values/policies when the model of the environment is not known.
>
> 6.   In section 3, we provide a subdifferential generalization of the Benveniste-Scheinkman theorem (Theorem D.2), which allows one to subdifferentiate the optimal value function associated with an MDP with parameterized rewards and action spaces. We use this theorem to derive optimality conditions for zero-sum stochastic Stackelberg games (Theorems 3.1 and D3), but this theorem may have other applications as well. These results can be then used to prove that the Stackelberg equilibrium of a particular zero-sum stochastic Stackelberg game corresponds to the solutions of a given optimization problem. For instance, in Theorem 4.1 we show that stochastic Fisher markets can be solved as zero-sum stochastic Stackelberg games.
>
> (You can find references in the following answer)

---

> > ### Author Response · Authors · 2022-08-01
> > **Response to Reviewer nfFs Continued and References**
> >
> >
> > 7. Yes, it is definitely possible to experiment with finite discrete-action spaces; we will consider adding some of our early experiments in such settings to the appendix. However, it is not clear whether [12] would be a good baseline, since these algorithms are formulated for static min-max Stackelberg games with independent strategy sets, i.e., where the actions of the leader does not affect the space of actions available to the follower. That said, we will be sure to add [12] to our related works section, since it is relevant to the literature on Stackelberg games.
> >
> > 8. Additional applications of our model include autonomous driving [13], reach-avoid problems in human-robot interaction [14], and robust optimization in stochastic environments [15]. These papers [13,14,15] use zero-sum stochastic Stackelberg games to model these applications, but the algorithms they use to solve their problems do not have any theoretical guarantees. Our work provides algorithms with theoretical guarantees for their problem settings.
> >
> > [1] Vorobeychik, Y., & Singh, S. (2012). Computing Stackelberg equilibria in discounted stochastic games. In Proceedings of the AAAI Conference on Artificial Intelligence (Vol. 26, No. 1, pp. 1478-1484).
> >
> > [2] Chang, Yanling, Alan L. Erera, and Chelsea C. White. "A leader–follower partially observed, multiobjective Markov game." Annals of Operations Research 235.1 (2015): 103-128.
> >
> > [3] Sengupta, S., & Kambhampati, S. (2020). Multi-agent reinforcement learning in Bayesian Stackelberg Markov games for adaptive moving target defense. arXiv preprint arXiv:2007.10457.
> >
> > [4] Ramponi, G., & Restelli, M. (2022, February). Learning in Markov Games: can we exploit a general-sum opponent?. In The 38th Conference on Uncertainty in Artificial Intelligence.
> >
> > [5] Vu, Q. L., Alumbaugh, Z., Ching, R., Ding, Q., Mahajan, A., Chasnov, B., ... & Ratliff, L. J. (2022, March). Stackelberg Policy Gradient: Evaluating the Performance of Leaders and Followers. In ICLR 2022 Workshop on Gamification and Multiagent Solutions.
> >
> > [6] Denizalp Goktas and Amy Greenwald. “Convex-Concave Min-Max Stackelberg Games”. In:351 Advances in Neural Information Processing Systems 34 (2021).
> >
> > [7] Denizalp Goktas and Amy Greenwald. Robust No-Regret Learning in Min-Max Stackelberg
> > Games. 2022.
> >
> > [8] Littman, M. L. (1994). Markov games as a framework for multi-agent reinforcement learning. In Machine learning proceedings 1994 (pp. 157-163). Morgan Kaufmann.
> >
> > [9] Bertsimas, Dimitris, and John Tsitsiklis. "Simulated annealing." Statistical science 8.1 (1993): 10-15.
> >
> > [10] Poli, Riccardo, James Kennedy, and Tim Blackwell. "Particle swarm optimization." Swarm intelligence 1.1 (2007): 33-57.
> >
> > [11] Bellman, Richard. "The theory of dynamic programming." Bulletin of the American Mathematical Society 60.6 (1954): 503-515.
> >
> > [12] Fiez, T., Chasnov, B., & Ratliff, L. (2020, November). Implicit learning dynamics in Stackelberg games: Equilibria characterization, convergence analysis, and empirical study. In International Conference on Machine Learning (pp. 3133-3144). PMLR.
> >
> > [13] Jaime F Fisac et al. “Reach-avoid problems with time-varying dynamics, targets and constraints”. In: Proceedings of the 18th international conference on hybrid systems: computation and control. 2015, pp. 11–20.
> >
> > [14] Somil Bansal et al. “Hamilton-jacobi reachability: A brief overview and recent advances”. In:
> > 2017 IEEE 56th Annual Conference on Decision and Control (CDC). IEEE. 2017, pp. 2242–358
> > 2253.
> >
> > [15] Dimitris Bertsimas, David B Brown, and Constantine Caramanis. “Theory and applications of robust optimization”. In: SIAM review 53.3 (2011), pp. 464–501.
> >
> > [16] Dempe, Stephan, and Alain Zemkoho. "Bilevel optimization." Springer optimization and its applications. Vol. 161. Springer Cham, 2020.

---

> > > ### Comment · Reviewer_nfFs · 2022-08-08
> > > **Answer to the authors**
> > >
> > > I would like to thank the authors for carefully answer to the raised questions. The answers clarify my doubts, however, I would like to ask them for the following clarification.
> > >
> > > In merits of related work, could the authors briefly comment on the relations with [1]? I think that the setting is different, but the two works seem connected. They showed a result that states that the leader's optimal policy can be not stationary. How does this result is reflected in your paper?
> > >
> > > In general, I think that the related work section needs to be revised.
> > >
> > >
> > > [1] Vorobeychik, Y., & Singh, S. (2012). Computing Stackelberg equilibria in discounted stochastic games (corrected version).

---

> > > > ### Author Response · Authors · 2022-08-09
> > > > **Response to Reviewer nfFs’s response**
> > > >
> > > > Vorobeychik and Singh consider a model which is both more general than ours, and more specific. It is more general in that they consider a general-sum stochastic Stackelberg game setting; it is more specific, in that the actions of the leader do not affect the feasible strategies of the follower. As a result, our findings do not contradict those of Vorobeychik and Singh.
> > > >
> > > > The set of Stackelberg equilibria in general-sum (one-shot) Stackelberg games is not convex (see for instance [3]), while in zero-sum (one-shot) Stackelberg games it is (Proposition B.3. of [2]). One can therefore construct an automorphism over the set of Stackelberg equilibria in the zero-sum case, whose fixed points correspond to recSE, which by the convexity of the domain of the automorphism are guaranteed to exist via Kakutani’s fixed point theorem. Such a fixed point is not guaranteed to exist in general-sum Stackelberg games because the domain of the automorphism is not convex.
> > > >
> > > > Note that our results provide an answer to the following question posed by Vorobeychik and Singh: “A natural question is whether there is any setting where a positive result is possible, besides zero-sum games, where there is no distinction between Nash equilibria and SSE” [1].
> > > >
> > > > [1] Vorobeychik, Y., & Singh, S. (2012). Computing Stackelberg equilibria in discounted stochastic games (corrected version).
> > > >
> > > > [2] Denizalp Goktas and Amy Greenwald. “Convex-Concave Min-Max Stackelberg Games”. In: 351 Advances in Neural Information Processing Systems 34 (2021).
> > > >
> > > > [3] Dempe, Stephan. Foundations of bilevel programming. Springer Science & Business Media, 2002.

---

### Meta-Review · Area_Chair_Q1o4 · 2022-08-27

**Recommendation:** Accept
**Confidence:** Certain

**Metareview:**

There was a positive consensus amongst the reviewers about this paper. It examines an interesting setting and makes worthwhile contributions. I believe that it that would make a nice contribution to NeurIPS and that it is likely to lead to more follow-up work. At the same time the authors should take care to improve the discussion of the related work and improve readability at the points that the reviewers have identified.

**Award:**

No

---

### Decision · Program_Chairs · 2022-09-14

Accept